# GENERALIZATION BOUNDS VIA DISTILLATION

**Daniel Hsu**
Columbia University, New York City
djhsu@cs.columbia.edu

**Ziwei Ji, Matus Telgarsky, Lan Wang**
University of Illinois, Urbana-Champaign
{ziweiji2,mjt,lanwang2}@illinois.edu

## ABSTRACT

This paper theoretically investigates the following empirical phenomenon: given a high-complexity network with poor generalization bounds, one can *distill* it into a network with nearly identical predictions but low complexity and vastly smaller generalization bounds. The main contribution is an analysis showing that the original network inherits this good generalization bound from its distillation, assuming the use of well-behaved data augmentation. This bound is presented both in an abstract and in a concrete form, the latter complemented by a reduction technique to handle modern computation graphs featuring convolutional layers, fully-connected layers, and skip connections, to name a few. To round out the story, a (looser) classical uniform convergence analysis of compression is also presented, as well as a variety of experiments on `cifar10` and `mnist` demonstrating similar generalization performance between the original network and its distillation.

## 1 OVERVIEW AND MAIN RESULTS

Generalization bounds are statistical tools which take as input various measurements of a predictor on training data, and output a performance estimate for unseen data — that is, they estimate how well the predictor *generalizes* to unseen data. Despite extensive development spanning many decades (Anthony & Bartlett, 1999), there is growing concern that these bounds are not only disastrously loose (Dziugaite & Roy, 2017), but worse that they do not correlate with the underlying phenomena (Jiang et al., 2019b), and even that the basic method of proof is doomed (Zhang et al., 2016; Nagarajan & Kolter, 2019). As an explicit demonstration of the looseness of these bounds, Figure 1 calculates bounds for a standard ResNet architecture achieving test errors of respectively 0.008 and 0.067 on `mnist` and `cifar10`; the observed generalization gap is $10^{-1}$, while standard generalization techniques upper bound it with $10^{15}$.

Contrary to this dilemma, there is evidence that these networks can often be compressed or *distilled* into simpler networks, while still preserving their output values and low test error. Meanwhile, these simpler networks exhibit vastly better generalization bounds: again referring to Figure 1, those same networks from before can be distilled with hardly any change to their outputs, while their bounds reduce by a factor of roughly $10^{10}$. Distillation is widely studied (Buciluǎ et al., 2006; Hinton et al., 2015), but usually the original network is discarded and only the final distilled network is preserved.

The purpose of this work is to carry the good generalization bounds of the distilled network back to the original network; in a sense, the explicit simplicity of the distilled network is used as a witness to implicit simplicity of the original network. The main contributions are as follows.

- The main theoretical contribution is a generalization bound for the original, undistilled network which scales primarily with the generalization properties of its distillation, assuming that well-behaved data augmentation is used to measure the *distillation distance*. An abstract version of this bound is stated in Lemma 1.1, along with a sufficient data augmentation technique in Lemma 1.2. A concrete version of the bound, suitable to handle the ResNet architecture in Figure 1, is described in Theorem 1.3. Handling sophisticated architectures with only minor proof alterations is another contribution of this work, and is described alongside Theorem 1.3. This abstract and concrete analysis is sketched in Section 3, with full proofs deferred to appendices.

- Rather than using an assumption on the distillation process (e.g., the aforementioned "well-behaved data augmentation"), this work also gives a direct uniform convergence analysis, culminating in Theorem 1.4. This is presented partially as an open problem or cautionary tale, as

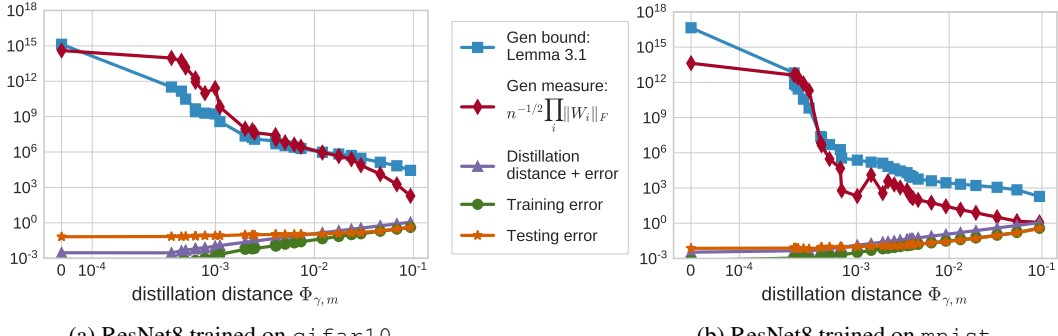

(a) ResNet8 trained on `cifar10`.     (b) ResNet8 trained on `mnist`.

Figure 1: **Generalization bounds throughout distillation.** These two subfigures track a sequence of increasingly distilled/compressed ResNet8 networks along their horizontal axes, respectively for `cifar10` and `mnist` data. This horizontal axis measures *distillation distance* $\Phi_{\gamma,m}$, as defined below in eq. (1.1). The bottom curves measure various training and testing errors, whereas the top two curves measure respectively a generalization bound presented here (cf. Theorem 1.3 and Lemma 3.1), and a *generalization measure*. Notably, the top two curves drop throughout a long interval during which test error remains small. For further experimental details, see Section 2.

- its proof is vastly more sophisticated than that of Theorem 1.3, but ultimately results in a much looser analysis. This analysis is sketched in Section 3, with full proofs deferred to appendices.

- While this work is primarily theoretical, it is motivated by Figure 1 and related experiments: Figures 2 to 4 demonstrate that not only does distillation improve generalization upper bounds, but moreover it makes them sufficiently tight to capture intrinsic properties of the predictors, for example removing the usual bad dependence on width in these bounds (cf. Figure 3). These experiments are detailed in Section 2.

## 1.1 AN ABSTRACT BOUND VIA DATA AUGMENTATION

This subsection describes the basic distillation setup and the core abstract bound based on data augmentation, culminating in Lemmas 1.1 and 1.2; a concrete bound follows in Section 1.2.

Given a multi-class predictor $f : \mathbb{R}^d \to \mathbb{R}^k$, distillation finds another predictor $g : \mathbb{R}^d \to \mathbb{R}^k$ which is simpler, but close in *distillation distance* $\Phi_{\gamma,m}$, meaning the *softmax outputs* $\phi_\gamma$ are close on average over a set of points $(z_i)_{i=1}^m$:

$$\Phi_{\gamma,m}(f,g) := \frac{1}{m}\sum_{i=1}^m \left\|\phi_\gamma(f(z_i)) - \phi_\gamma(g(z_i))\right\|_1, \quad \text{where } \phi_\gamma(f(z)) \propto \exp\left(f(z)/\gamma\right). \quad (1.1)$$

The quantity $\gamma > 0$ is sometimes called a *temperature* (Hinton et al., 2015). Decreasing $\gamma$ increases sensitivity near the decision boundary; in this way, it is naturally related to the concept of *margins* in generalization theory, as detailed in Appendix B. due to these connections, the use of softmax is beneficial in this work, though not completely standard in the literature (Buciluǔ et al., 2006).

We can now outline Figure 1 and the associated empirical phenomenon which motivates this work. (Please see Section 2 for further details on these experiments.) Consider a predictor $f$ which has good test error but bad generalization bounds; by treating the distillation distance $\Phi_{\gamma,m}(f,g)$ as an objective function and increasingly regularizing $g$, we obtain a sequence of predictors $(g_0, \ldots, g_t)$, where $g_0 = f$, which trade off between distillation distance and predictor complexity. The curves in Figure 1 are produced in exactly this way, and demonstrate that there are predictors nearly identical to the original $f$ which have vastly smaller generalization bounds.

Our goal here is to show that this is enough to imply that $f$ in turn must *also* have good generalization bounds, despite its apparent complexity. To sketch the idea, by a bit of algebra (cf. Lemma A.2), we can upper bound error probabilities with *expected* distillation distances and errors:

$$\Pr_{x,y}[\arg\max_{y'} f(x)_{y'} \neq y] \leq 2\mathbb{E}_x\left\|\phi_\gamma(f(x)) - \phi_\gamma(g(x))\right\|_1 + 2\mathbb{E}_{x,y}\left(1 - \phi_\gamma(g(x))_y\right).$$

The next step is to convert these expected errors into quantities over the training set. The last term is already in a form we want: it depends only on $g$, so we can apply uniform convergence with the low complexity of $g$. (Measured over the training set, this term is the *distillation error* in Figure 1.)

The expected distillation distance term is problematic, however. Here are two approaches.

1. We can directly apply uniform convergence; for instance, this approach was followed by Suzuki et al. (2019), and a more direct approach is followed here to prove Theorem 1.4. Unfortunately, it is unclear how this technique can avoid paying significantly for the high complexity of $f$.

2. The idea in this subsection is to somehow trade off computation for the high statistical cost of the complexity of $f$. Specifically, notice that $\Phi_{\gamma,m}(f,g)$ only relies upon the marginal distribution of the inputs $x$, and not their labels. This subsection will pay computation to estimate $\Phi_{\gamma,m}$ with extra samples via *data augmentation*, offsetting the high complexity of $f$.

We can now set up and state our main distillation bound. Suppose we have a training set $((x_i, y_i))_{i=1}^n$ drawn from some measure $\mu$, with marginal distribution $\mu_{\mathcal{X}}$ on the inputs $x$. Suppose we also have $(z_i)_{i=1}^m$ drawn from a *data augmentation distribution* $\nu_n$, the subscript referring to the fact that it depends on $(x_i)_{i=1}^n$. Our analysis works when $\|\mathrm{d}\mu_{\mathcal{X}}/\mathrm{d}\nu_n\|_\infty$, the ratio between the two densities, is finite. If it is large, then one can tighten the bound by sampling more from $\nu_n$, which is a computational burden; explicit bounds on this term will be given shortly in Lemma 1.2.

**Lemma 1.1.** *Let temperature parameter $\gamma > 0$ be given, along with sets of multiclass predictors $\mathcal{F}$ and $\mathcal{G}$. Then with probability at least $1 - 2\delta$ over an iid draw of data $((x_i, y_i))_{i=1}^n$ from $\mu$ and $(z_i)_{i=1}^m$ from $\nu_n$, every $f \in \mathcal{F}$ and $g \in \mathcal{G}$ satisfy*

$$\Pr[\arg\max_{y'} f(x)_{y'} \neq y] \leq 2\left\|\frac{\mathrm{d}\mu_{\mathcal{X}}}{\mathrm{d}\nu_n}\right\|_\infty \Phi_{\gamma,m}(f,g) + \frac{2}{n}\sum_{i=1}^n \left(1 - \phi_\gamma(g(x_i))_{y_i}\right)$$

$$+ \widetilde{\mathcal{O}}\left(\frac{k^{3/2}}{\gamma}\left\|\frac{\mathrm{d}\mu_{\mathcal{X}}}{\mathrm{d}\nu_n}\right\|_\infty \left(\mathrm{Rad}_m(\mathcal{F}) + \mathrm{Rad}_m(\mathcal{G})\right) + \frac{\sqrt{k}}{\gamma}\mathrm{Rad}_n(\mathcal{G})\right)$$

$$+ 6\sqrt{\frac{\ln(1/\delta)}{2n}}\left(1 + \left\|\frac{\mathrm{d}\mu_{\mathcal{X}}}{\mathrm{d}\nu_n}\right\|_\infty \sqrt{\frac{n}{m}}\right),$$

*where Rademacher complexities $\mathrm{Rad}_n$ and $\mathrm{Rad}_m$ are defined in Section 1.4.*

A key point is that the Rademacher complexity $\mathrm{Rad}_m(\mathcal{F})$ of the complicated functions $\mathcal{F}$ has a subscript "$m$", which explicitly introduces a factor $1/m$ in the complexity definition (cf. Section 1.4). As such, sampling more from the data augmentation measure can mitigate this term, and leave the complexity of the distillation class $\mathcal{G}$ as the dominant term.

Of course, this also requires $\|\mathrm{d}\mu_{\mathcal{X}}/\mathrm{d}\nu_n\|_\infty$ to be reasonable. As follows is one data augmentation scheme (and assumption on marginal distribution $\mu_{\mathcal{X}}$) which ensures this.

**Lemma 1.2.** *Let $(x_i)_{i=1}^n$ be a data sample drawn iid from $\mu_{\mathcal{X}}$, and suppose the corresponding density $p$ is supported on $[0,1]^d$ and is Hölder continuous, meaning $|p(x) - p(x')| \leq C_\alpha \|x - x'\|^\alpha$ for some $C_\alpha \geq 0, \alpha \in [0,1]$. Define a data augmentation measure $\nu_n$ via the following sampling procedure.*

- *With probability $1/2$, sample $z$ uniformly within $[0,1]^d$.*

- *Otherwise, select a data index $i \in [n]$ uniformly, and sample $z$ from a Gaussian centered at $x_i$, and having covariance $\sigma^2 I$ where $\sigma := n^{-1/(2\alpha+d)}$.*

*Then with probability at least $1 - 1/n$ over the draw of $(x_i)_{i=1}^n$,*

$$\left\|\frac{\mathrm{d}\mu_{\mathcal{X}}}{\mathrm{d}\nu_n}\right\|_\infty = 4 + \mathcal{O}\left(\frac{\sqrt{\ln n}}{n^{\alpha/(2\alpha+d)}}\right).$$

Though the idea is not pursued here, there are other ways to control $\|\mathrm{d}\mu_{\mathcal{X}}/\mathrm{d}\nu_n\|_\infty$, for instance via an independent sample of unlabeled data; Lemma 1.1 is agnostic to these choices.

## 1.2 A CONCRETE BOUND FOR COMPUTATION GRAPHS

This subsection gives an explicit complexity bound which starts from Lemma 1.1, but bounds $\|\mathrm{d}\mu_\mathcal{X}/\mathrm{d}\nu_n\|_\infty$ via Lemma 1.2, and also includes an upper bound on Rademacher complexity which can handle the ResNet, as in Figure 1. A side contribution of this work is the formalism to easily handle these architectures, detailed as follows.

*Canonical computation graphs* are a way to write down feedforward networks which include dense linear layers, convolutional layers, skip connections, and multivariate gates, to name a few, all while allowing the analysis to look roughly like a regular dense network. The construction applies directly to batches: given an input batch $X \in \mathbb{R}^{n \times d}$, the output $X_i$ of layer $i$ is defined inductively as

$$
X_0^\mathsf{T} := X^\mathsf{T}, \qquad X_i^\mathsf{T} := \sigma_i\left([W_i \Pi_i D_i \rangle F_i] X_{i-1}^\mathsf{T}\right) = \sigma_i\left(\begin{bmatrix} W_i \Pi_i D_i X_{i-1}^\mathsf{T} \\ F_i X_{i-1}^\mathsf{T} \end{bmatrix}\right),
$$

where: $\sigma_i$ is a multivariate-to-multivariate $\rho_i$-Lipschitz function (measured over minibatches on either side with Frobenius norm); $F_i$ is a *fixed* matrix, for instance an identity mapping as in a residual network's skip connection; $D_i$ is a *fixed* diagonal matrix selecting certain coordinates, for instance the non-skip part in a residual network; $\Pi_i$ is a Frobenius norm projection of a full minibatch; $W_i$ is a weight matrix, the trainable parameters; $[W_i \Pi_i D_i \rangle F_i]$ denotes *row-wise concatenation* of $W_i \Pi_i D_i$ and $F_i$.

As a simple example of this architecture, a multi-layer skip connection can be modeled by including identity mappings in all relevant fixed matrices $F_i$, and also including identity mappings in the corresponding coordinates of the multivariate gates $\sigma_i$. As a second example, note how to model convolution layers: each layer outputs a matrix whose rows correspond to examples, but nothing prevents the batch size from changes between layers; in particular, the multivariate activation before a convolution layer can reshape its output to have each row correspond to a patch of an input image, whereby the convolution filter is now a regular dense weight matrix.

A fixed computation graph architecture $\mathcal{G}(\vec{\rho}, \vec{b}, \vec{r}, \vec{s})$ has associated hyperparameters $(\vec{\rho}, \vec{b}, \vec{r}, \vec{s})$, described as follows. $\vec{\rho}$ is the set of Lipschitz constants for each (multivariate) gate, as described before. $r_i$ is a norm bound $\|W_i^\mathsf{T}\|_{2,1} \leq r_i$ (sum of the $\|\cdot\|_2$-norms of the rows), $b_i\sqrt{n}$ (where $n$ is the input batch size) is the radius of the Frobenius norm ball which $\Pi_i$ is projecting onto, and $s_i$ is the operator norm of $X \mapsto [W_i \Pi_i D_i X^\mathsf{T} \rangle F_i X^\mathsf{T}]$. While the definition is intricate, it cannot only model basic residual networks, but it is sensitive enough to be able to have $s_i = 1$ and $r_i = 0$ when residual blocks are fully zeroed out, an effect which indeed occurs during distillation.

**Theorem 1.3.** *Let temperature parameter $\gamma > 0$ be given, along with multiclass predictors $\mathcal{F}$, and a computation graph architecture $\mathcal{G}$. Then with probability at least $1 - 2\delta$ over an iid draw of data $((x_i, y_i))_{i=1}^n$ from $\mu$ and $(z_i)_{i=1}^n$ from $\nu_n$, every $f \in \mathcal{F}$ satisfies*

$$
\Pr[\arg\max_{y'} f(x)_{y'} \neq y] \leq \inf_{\substack{(\vec{b},\vec{r},\vec{s}) \geq 1 \\ g \in \mathcal{G}(\vec{\rho},\vec{b},\vec{r},\vec{s})}} 2\left[\left\|\frac{\mathrm{d}\mu_\mathcal{X}}{\mathrm{d}\nu_n}\right\|_\infty \Phi_{\gamma,m}(f,g) + \frac{2}{n}\sum_{i=1}^n (1 - \phi_\gamma(g(x_i))_{y_i}\right.
$$

$$
+ \widetilde{\mathcal{O}}\left(\frac{k^{3/2}}{\gamma}\left\|\frac{\mathrm{d}\mu_\mathcal{X}}{\mathrm{d}\nu_n}\right\|_\infty \mathrm{Rad}_m(\mathcal{F})\right) + 6\sqrt{\frac{\ln(1/\delta)}{2n}}\left(1 + \left\|\frac{\mathrm{d}\mu_\mathcal{X}}{\mathrm{d}\nu_n}\right\|_\infty \sqrt{\frac{n}{m}}\right)
$$

$$
\left. + \widetilde{\mathcal{O}}\left(\frac{\sqrt{k}}{\gamma\sqrt{n}}\left(1 + k\left\|\frac{\mathrm{d}\mu_\mathcal{X}}{\mathrm{d}\nu_n}\right\|_\infty \sqrt{\frac{n}{m}}\right)\left(\sum_i\left[r_i b_i \rho_i \prod_{l=i+1}^L s_l \rho_l\right]^{2/3}\right)^{3/2}\right)\right].
$$

*Under the conditions of Lemma 1.2, ignoring an additional failure probability $1/n$, then $\|\frac{\mathrm{d}\mu_\mathcal{X}}{\mathrm{d}\nu_n}\|_\infty = 4 + \mathcal{O}\left(\frac{\sqrt{\ln n}}{n^{\alpha/(2\alpha+d)}}\right)$.*

A proof sketch of this bound appears in Section 3, with full details deferred to appendices. The proof is a simplification of the covering number argument from (Bartlett et al., 2017a); for another computation graph formalism designed to work with the covering number arguments from (Bartlett et al., 2017a), see the generalization bounds due to Wei & Ma (2019).

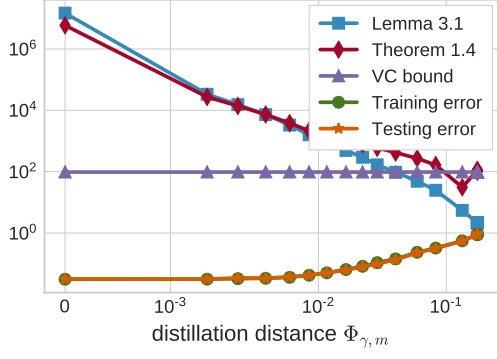
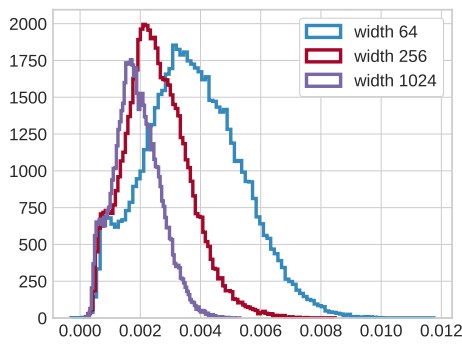

(a) Comparison of bounds on `cifar10`.    (b) Width dependence with Theorem 1.4.

Figure 2: **Performance of stable rank bound (cf. Theorem 1.4).** Figure 2a compares Theorem 1.4 to Lemma 3.1 and the VC bound (Bartlett et al., 2017b), and Figure 2b normalizes the *margin histogram* by Theorem 1.4, showing an unfortunate failure of width independence (cf. Figure 3). For details and a discussion of margin histograms, see Section 2.

### 1.3 A UNIFORM-CONVERGENCE APPROACH TO DISTILLATION

In this section, we derive a Rademacher complexity bound on $\mathcal{F}$ whose proof *internally* uses compression; specifically, it first replaces $f$ with a narrower network $g$, and then uses a covering number bound sensitive to network size to control $g$. The proof analytically chooses $g$'s width based on the structure of $f$ and also the provided data, and this data dependence incurs a factor which causes the familiar $1/\sqrt{n}$ rate to worsen to $1/n^{1/4}$ (which appears as $\|X\|_{\mathrm{F}}/n^{3/4}$). This proof is much more intricate than the proofs coming before, and cannot handle general computation graphs, and also ignores the beneficial structure of the softmax.

**Theorem 1.4.** *Let data matrix $X \in \mathbb{R}^{n \times d}$ be given, and let $\mathcal{F}$ denote networks of the form $x \mapsto \sigma_L(W_L \cdots \sigma_1(W_1 x))$ with spectral norm $\|W_i\|_2 \leq s_i$, and 1-Lipschitz and 1-homogeneous activations $\sigma_i$, and $\|W_i\|_{\mathrm{F}} \leq R_i$ and width at most $m$. Then*

$$\mathrm{Rad}(\mathcal{F}) = \widetilde{\mathcal{O}}\left(\frac{\|X\|_{\mathrm{F}}}{n^{3/4}}\left[\prod_j s_j\right]\left[\sum_i (R_i/s_i)^{4/5}\right]^{5/4}\left[\sum_i \ln R_i\right]^{1/4}\right).$$

The term $R_i/s_i$ is the square root of the *stable rank* of weight matrix $W_i$, and is a desirable quantity in a generalization bound: it scales more mildly with width than terms like $\|W_i^{\mathsf{T}}\|_{2,1}$ and $\|W_i^{\mathsf{T}}\|_{\mathrm{F}}\sqrt{\text{width}}$ which often appear (the former appears in Theorem 1.3 and Lemma 3.1). Another stable rank bound was developed by Suzuki et al. (2019), but has an extra mild dependence on width.

As depicted in Figure 2, however, this bound is not fully width-independent. Moreover, we can compare it to Lemma 3.1 throughout distillation, and not only does this bound not capture the power of distillation, but also, eventually its bad dependence on $n$ causes it to lose out to Lemma 3.1.

### 1.4 ADDITIONAL NOTATION

Given data $(z_i)_{i=1}^n$, the *Rademacher complexity* of univariate functions $\mathcal{H}$ is

$$\mathrm{Rad}(\mathcal{H}) := \mathbb{E}_{\vec{\epsilon}} \sup_{h \in \mathcal{H}} \frac{1}{n} \sum_i \epsilon_i h(z_i), \qquad \text{where } \epsilon_i \overset{\text{i.i.d.}}{\sim} \text{Uniform}(\{-1, +1\}).$$

Rademacher complexity is the most common tool in generalization theory (Shalev-Shwartz & Ben-David, 2014), and is incorporated in Lemma 1.1 due to its convenience and wide use. To handle multivariate (multiclass) outputs, the definition is overloaded via the worst case labels as $\mathrm{Rad}_n(\mathcal{F}) = \sup_{\vec{y} \in [k]^n} \mathrm{Rad}(\{(x, y) \mapsto f(x)_y : f \in \mathcal{F}\})$. This definition is for mathematical convenience, but overall not ideal; Rademacher complexity seems to have difficulty dealing with such geometries.

Regarding norms, $\|\cdot\| = \|\cdot\|_{\mathrm{F}}$ will denote the Frobenius norm, and $\|\cdot\|_2$ will denote spectral norm.

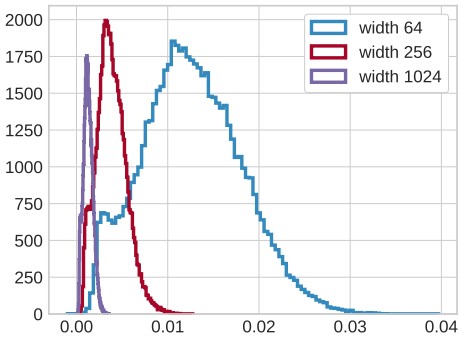 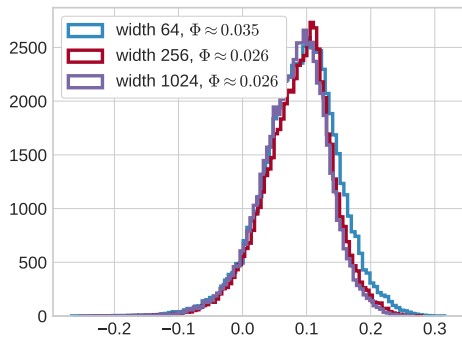

(a) Margins before distillation.   (b) Margins after distillation.

Figure 3: **Width independence.** Fully-connected 6-layer networks of widths $\{64, 256, 1024\}$ were trained on `mnist` until training error zero; the margin histograms, normalized by the generalization bound in Lemma 3.1, all differ, and are close to zero. After distillation, the margin distributions are far from zero and nearly the same. In the distillation legend, the second term $\Phi_{\gamma,m}$ denotes the distillation distance, as defined in Equation (1.1). Experiment details and an explanation of margin histograms appear in Section 2.

## 2 ILLUSTRATIVE EMPIRICAL RESULTS

This section describes the experimental setup, and the main experiments: Figure 1 showing progressive distillation, Figure 2 comparing Theorem 1.4, Lemma 3.1 and VC dimension, Figure 3 showing width independence after distillation, and Figure 4 showing the effect of random labels.

**Experimental setup.** As sketched before, networks were trained in a standard way on either `cifar10` or `mnist`, and then *distilled* by trading off between complexity and distillation distance $\Phi_{\gamma,m}$. Details are as follows.

1. **Training initial network $f$.** In Figures 1 and 2a, the architecture was a ResNet8 based on one used in (Coleman et al., 2017), and achieved test errors 0.067 and 0.008 on `cifar10` and `mnist`, respectively, with no changes to the setup and a modest amount of training; the training algorithm was Adam; this and most other choices followed the scheme in (Coleman et al., 2017) to achieve a competitively low test error on `cifar10`. In Figures 2b, 3 and 4, a 6-layer fully connected network was used (width 8192 in Figure 2b, widths $\{64, 256, 1024\}$ in Figure 3, width 256 in Figure 4), and vanilla SGD was used to optimize.

2. **Training distillation network $g$.** Given $f$ and a regularization strength $\lambda_j$, each distillation $g_j$ was found via approximate minimization of the objective

$$g \mapsto \Phi_{\gamma,m}(f, g) + \lambda_j \text{Complexity(g)}. \tag{2.1}$$

   In more detail, first $g_0$ was initialized to $f$ ($g$ and $f$ always used the same architecture) and optimized via eq. (2.1) with $\lambda_0$ set to roughly $\text{risk}(f)/\text{Complexity}(f)$, and thereafter $g_{j+1}$ was initialized to $g_j$ and found by optimizing eq. (2.1) with $\lambda_{j+1} := 2\lambda_j$. The optimization method was the same as the one used to find $f$. The term Complexity$(g)$ was some computationally reasonable approximation of Lemma 3.1: for Figures 2b, 3 and 4, it was just $\sum_i \|W_i^\mathsf{T}\|_{2,1}$, but for Figures 1 and 2a, it also included a tractable surrogate for the product of the spectral norms, which greatly helped distillation performance with these deeper architectures.

   In Figures 2b, 3 and 4, a full regularization sequence was not shown, only a single $g_j$. This was chosen with a simple heuristic: amongst all $(g_j)_{j \geq 1}$, pick the one whose 10% margin quantile is largest (see the definition and discussion of margins below).

**Margin histograms.** Figures 2b, 3 and 4 all depict *margin histograms*, a flexible tool to study the individual predictions of a network on all examples in a training set (see for instance (Schapire & Freund, 2012) for their use studying boosting, and (Bartlett et al., 2017a; Jiang et al., 2019a) for

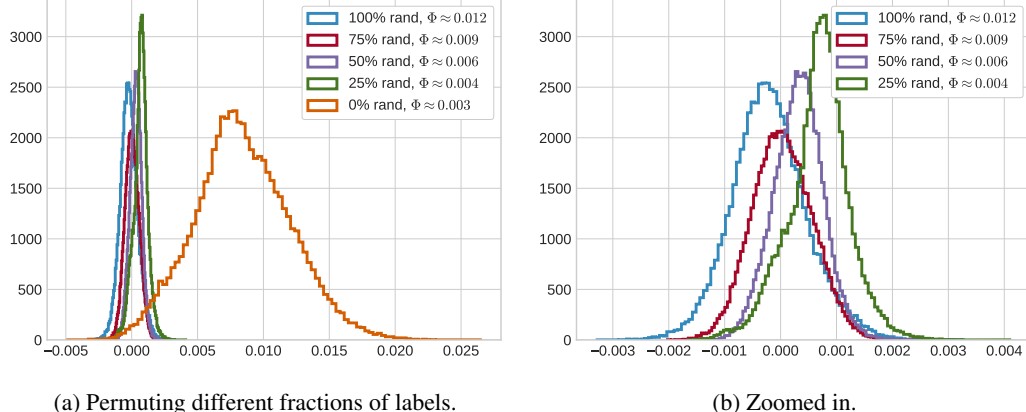

(a) Permuting different fractions of labels.  (b) Zoomed in.

Figure 4: **Label randomization.** Here $\{0\%, 25\%, 50\%, 75\%, 100\%\}$ of the labels were permuted across the respective experiments. In all cases, the margin distribution is collapsed to zero. For details, including an explanation of margin histograms, see Section 2.

their use in studying deep networks). Concretely, given a predictor $g \in \mathcal{G}$, the prediction on every example is replaced with a real scalar called the *normalized margin* via

$$(x_i, y_i) \mapsto \frac{g(x_i)_{y_i} - \max_{j \neq y_i} g(x_i)_j}{\mathrm{Rad}_n(\mathcal{G})},$$

where $\mathrm{Rad}_n(\mathcal{G})$ is the Rademacher complexity (cf. Section 1.4), and then the histogram of these $n$ scalars is plotted, with the horizontal axis values thus corresponding to normalized margins. By using Rademacher complexity as normalization, these margin distributions can be compared across predictors and even data sets, and give a more fine-grained analysis of the quality of the generalization bound. This normalization choice was first studied in (Bartlett et al., 2017a), where it was also mentioned that this normalization allows one to read off generalization bounds from the plot. Here, it also suggests reasonable values for the softmax temperature $\gamma$.

**Figure 1: effect of distillation on generalization bounds.** This figure was described before; briefly, a highlight is that in the initial phase, training and testing errors hardly change while bounds drop by a factor of nearly $10^{10}$. Regarding "generalization measure", this term appears in studies of quantities which correlate with generalization, but are not necessarily rigorous generalization bounds (Jiang et al., 2019b; Dziugaite et al., 2020); in this specific case, the product of Frobenius norms requires a dense ReLU network (Golowich et al., 2018), and is invalid for the ResNet (e.g., a complicated ResNet with a single identity residual block yields a value 0 by this measure).

**Figure 2a: comparison of Theorem 1.4, Lemma 3.1 and VC bounds.** Theorem 1.4 was intended to internalize distillation, but as in Figure 2a, clearly a subsequent distillation still greatly reduces the bound. While initially the bound is better than Lemma 3.1 (which does not internalize distillation), eventually the $n^{1/4}$ factor causes it to lose out. Also note that eventually the bounds beat the VC bound, which has been identified as a surprisingly challenging baseline (Arora et al., 2018).

**Figure 3: width independence.** Prior work has identified that generalization bounds are quite bad at handling changes in width, even if predictions and test error don't change much (Nagarajan & Kolter, 2019; Jiang et al., 2019b; Dziugaite et al., 2020). This is captured in Figure 3a, where the *margin distributions* (see above) with different widths are all very different, despite similar test errors. However, following distillation, the margin histograms in Figure 3b are nearly identical! That is to say: distillation not only decreases loose upper bounds as before, it tightens them to the point where they capture intrinsic properties of the predictors.

**Figure 2b: failure of width independence with Theorem 1.4.** The bound in Theorem 1.4 was designed to internalize compression, and there was some hope of this due to the stable rank term.

Unfortunately, Figure 2b shows that it doesn't quite succeed: while the margin histograms are less separated than for the undistilled networks in Figure 3a, they are still visibly separated unlike the post-distillation histograms in Figure 3b.

**Figure 4: random labels.** A standard sanity check for generalization bounds is whether they can reflect the difficulty of fitting random labels (Zhang et al., 2016). While it has been empirically shown that Rademacher bounds do sharply reflect the presence of random labels (Bartlett et al., 2017a, Figures 2 & 3), the effect is amplified with distillation: even randomizing just 25% shrinks the margin distribution significantly.

## 3 ANALYSIS OVERVIEW AND SKETCH OF PROOFS

This section sketches all proofs, and provides further context and connections to the literature. Full proof details appear in the appendices.

### 3.1 ABSTRACT DATA AUGMENTATION BOUNDS IN SECTION 1.1

As mentioned in Section 1.1, the first step of the proof is to apply Lemma A.2 to obtain

$$\Pr_{x,y}[\arg\max_{y'} f(x)_{y'} \neq y] \leq 2\mathbb{E}_x \left\| \phi_\gamma(f(x)) - \phi_\gamma(g(x)) \right\|_1 + 2\mathbb{E}_{x,y} \left( 1 - \phi_\gamma(g(x))_y \right);$$

this step is similar to how the ramp loss is used with margin-based generalization bounds, a connection which is discussed in Appendix B.

Section 1.1 also mentioned that the last term is easy: $\phi_\gamma$ is $(1/\gamma)$-Lipschitz, and we can peel it off and only pay the Rademacher complexity associated with $g \in \mathcal{G}$.

With data augmentation, the first term is also easy:

$$\mathbb{E}\Phi_{\gamma,m}(f,g) = \int \|\phi_\gamma(f(z)) - \phi_\gamma(g(z))\|_1 \, d\mu_{\mathcal{X}}(z) = \int \|\phi_\gamma(f(z)) - \phi_\gamma(g(z))\|_1 \frac{d\mu_{\mathcal{X}}}{d\nu_n} \, d\nu_n(z)$$

$$\leq \left\| \frac{d\mu_{\mathcal{X}}}{d\nu_n} \right\|_\infty \int \|\phi_\gamma(f(z)) - \phi_\gamma(g(z))\|_1 \, d\nu_n(z),$$

and now we may apply uniform convergence to $\nu_n$ rather than $\mu_{\mathcal{X}}$. In the appendix, this proof is handled with a bit more generality, allowing arbitrary norms, which may help in certain settings. All together, this leads to a proof of Lemma 1.1.

For the explicit data augmentation estimate in Lemma 1.2, the proof breaks into roughly two cases: low density regions where the uniform sampling gives the bound, and high density regions where the Gaussian sampling gives the bound. In the latter case, the Gaussian sampling in expectation behaves as a kernel density estimate, and the proof invokes a standard bound (Jiang, 2017).

### 3.2 CONCRETE DATA AUGMENTATION BOUNDS IN SECTION 1.2

The main work in this proof is the following generalization bound for computation graphs, which follows the proof scheme from (Bartlett et al., 2017a), though simplified in various ways, owing mainly to the omission of general matrix norm penalties on weight matrices, and the omission of the *reference matrices*. The reference matrices were a technique to center the weight norm balls away from the origin; a logical place to center them was at initialization. However, in this distillation setting, it is in fact most natural to center everything at the origin, and apply regularization and shrink to a well-behaved function (rather than shrinking back to the random initialization, which after all defines a complicated function). The proof also features a simplified $(2,1)$-norm matrix covering proof (cf. Lemma C.3).

**Lemma 3.1.** *Let data $X \in \mathbb{R}^{n \times d}$ be given. Let computation graph $\mathcal{G}$ be given, where $\Pi_i$ projects to Frobenius-norm balls of radius $b_i\sqrt{n}$, and $\|W_i^\intercal\|_{2,1} \leq r_i$, and $\|[W_i\Pi_i D_i]F_i]\|_2 \leq s_i$, and $\mathrm{Lip}(\sigma_i) \leq \rho_i$, and all layers have width at most $m$. Then for every $\epsilon > 0$ there exists a covering set $\mathcal{M}$ satisfying*

$$\sup_{g \in \mathcal{G}} \min_{\hat{X} \in \mathcal{M}} \left\| g(X^\intercal) - \hat{X} \right\| \leq \epsilon \quad \text{and} \quad \ln|\mathcal{M}| \leq \frac{2^{4/3} n \ln(2m^2)}{\epsilon^2} \left[ \sum_i \left( r_i b_i \rho_i \prod_{l=i+1}^{L} s_l \rho_l \right)^{2/3} \right]^3.$$

*Consequently,*

$$\text{Rad}(\mathcal{G}) \leq \frac{4}{n} + 12\sqrt{\frac{\ln(2m^2)}{n}} \left[ \sum_i \left( r_i b_i \rho_i \prod_{l=i+1}^{L} s_l \rho_l \right)^{2/3} \right]^{3/2}.$$

From there, the proof of Theorem 1.3 follows via Lemmas 1.1 and 1.2, and many union bounds.

### 3.3 DIRECT UNIFORM CONVERGENCE APPROACH IN THEOREM 1.4

As mentioned before, the first step of the proof is to sparsify the network, specifically each matrix product. Concretely, given weights $W_i$ of layer $i$, letting $X_{i-1}^\intercal$ denote the input to this layer, then

$$W_i X_{i-1}^\intercal = \sum_{j=1}^{m} (W_i \mathbf{e}_j)(X_{i-1}\mathbf{e}_j)^\intercal.$$

Written this way, it seems natural that the matrix product should "concentrate", and that considering all $m$ outer products should not be necessary. Indeed, exactly such an approach has been followed before to analyze randomized matrix multiplication schemes (Sarlos, 2006). As there is no goal of high probability here, the analysis is simpler, and follows from the Maurey lemma (cf. Lemma C.1), as is used in the $(2,1)$-norm matrix covering bound in Lemma C.3.

**Lemma 3.2.** *Let a network be given with 1-Lipschitz homogeneous activations $\sigma_i$ and weight matrices $(W_1, \ldots, W_L)$ of maximum width $m$, along with data matrix $X \in \mathbb{R}^{n \times d}$ and desired widths $(k_1, \ldots, k_L)$ be given. Then there exists a sparsified network output, recursively defined via*

$$\hat{X}_0^\intercal := X^\intercal, \quad \text{and} \quad \hat{X}_i^\intercal := \Pi_i \sigma_i(W_i M_i X_{i-1}^\intercal), \quad \text{where} \quad M_i := \sum_{j \in S_i} \frac{Z_j \mathbf{e}_j \mathbf{e}_j^\intercal}{\|A\mathbf{e}_j\|},$$

*where $S_i$ is a multiset of $k_i = |S_i|$ indices, $\Pi_i$ denotes projection onto the Frobenius-norm ball of radius $\|X\|_{\text{F}} \prod_{j \leq i} \|W_j\|_2$, and the scaling term $Z_j$ satisfies $Z_j \leq \|W_k\|_{\text{F}}\sqrt{m/k_j}$, and*

$$\|\sigma_L(W_L \cdots \sigma_1(W_1 X^\intercal) \cdots) - \hat{X}_L^\intercal\|_{\text{F}} \leq \|X\|_{\text{F}} \left[ \prod_{i=1}^{L} \|W_i\|_2 \right] \sum_{i=1}^{L} \sqrt{\frac{\|W_i\|_{\text{F}}^2}{k_i \|W_i\|_2^2}},$$

The statement of this lemma is lengthy and detailed because the exact guts of the construction are needed in the subsequent generalization proof. Specifically, now that there are few nodes, a generalization bound sensitive to narrow networks can be applied. On the surface, it seems reasonable to apply a VC bound, but this approach did not yield a rate better than $n^{-1/6}$, and also had an explicit dependence on the depth of the network, times other terms visible in Theorem 1.4.

Instead, the approach here, aiming for a better dependence on $n$ and also no explicit dependence on network depth, was to produce an $\infty$-norm covering number bound (see (Long & Sedghi, 2019) for a related approach), with some minor adjustments (indeed, the $\infty$-norm parameter covering approach was applied to obtain a Frobenius-norm bound, as in Lemma 3.1). Unfortunately, the magnitudes of weight matrix entries must be controlled for this to work (unlike the VC approach), and this necessitated the detailed form of Lemma 3.2 above.

To close with a few pointers to the literature, as Lemma 3.2 is essentially a pruning bound, it is potentially of independent interest; see for instance the literature on lottery tickets and pruning (Frankle & Carbin, 2019; Frankle et al., 2020; Su et al., 2020). Secondly, there is already one generalization bound in the literature which exhibits spectral norms, due to (Suzuki et al., 2019); unfortunately, it also has an explicit dependence on network width.

### ACKNOWLEDGMENTS

MT thanks Vaishnavh Nagarajan for helpful discussions and suggestions. ZJ and MT are grateful for support from the NSF under grant IIS-1750051, and from NVIDIA under a GPU grant.

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

## A    PROOFS FOR SECTION 1.1

The first step is an abstract version of Lemma 1.1 which does not explicitly involve the softmax, just bounded functions.

**Lemma A.1.** *Let classes of bounded functions $\mathcal{F}$ and $\mathcal{G}$ be given with $\mathcal{F} \ni f : \mathcal{X} \to [0,1]^k$ and $\mathcal{G} \ni g : \mathcal{X} \to [0,1]^k$. Let conjugate exponents $1/p + 1/q = 1$ be given. Then with probability at least $1 - 2\delta$ over the draw of $((x_i, y_i))_{i=1}^n$ from $\mu$ and $(z_i)_{i=1}^m$ from $\nu_n$, for every $f \in \mathcal{F}$ and $g \in \mathcal{G}$,*

$$\mathbb{E}f(x)_y \leq \frac{1}{n}\sum_{i=1}^n g(x_i)_{y_i} + 2\mathrm{Rad}_n\left(\{(x,y) \mapsto g(x)_y : g \in \mathcal{G}\}\right) + 3\sqrt{\frac{\ln(1/\delta)}{2n}}$$

$$+ \left\|\frac{\mathrm{d}\mu_{\mathcal{X}}}{\mathrm{d}\nu_n}\right\|_{L_q(\nu_n)} \left(\frac{1}{m}\sum_{i=1}^m \|f(z_i) - g(z_i)\|_p^p + 3\sqrt{\frac{\ln(1/\delta)}{2m}}\right.$$

$$\left. + 2\mathrm{Rad}_m\left(\left\{z \mapsto \min\{1, \|f(z) - g(z)\|_p^p\} : f \in \mathcal{F}, g \in \mathcal{G}\right\}\right)\right)^{1/p}$$

*where*

$$\mathrm{Rad}_m\left(\left\{z \mapsto \min\{1, \|f(z) - g(z)\|_p^p\} : f \in \mathcal{F}, g \in \mathcal{G}\right\}\right)$$

$$\leq p \sum_{y'=1}^k \left[\mathrm{Rad}_m(\{z \mapsto f(z)_{y'} : f \in \mathcal{F}\}) + \mathrm{Rad}_m(\{z \mapsto g(z)_{y'} : g \in \mathcal{G}\})\right].$$

*Proof of Lemma A.1.* To start, for any $f \in \mathcal{F}$ and $g \in \mathcal{G}$, write

$$\mathbb{E}f(x)_y = \mathbb{E}(f(x) - g(x))_y + \mathbb{E}g(x)_y.$$

The last term is easiest, and let's handle it first: by standard Rademacher complexity arguments (Shalev-Shwartz & Ben-David, 2014), with probability at least $1 - \delta$, every $g \in \mathcal{G}$ satisfies

$$\mathbb{E}g(x)_y \leq \frac{1}{n}\sum_{i=1}^n g(x_i)_{y_i} + 2\mathrm{Rad}_n(\{(x,y) \mapsto g(x)_y : g \in \mathcal{G}\}) + 3\sqrt{\frac{\ln(1/\delta)}{2n}}.$$

For the first term, since $f : \mathcal{X} \to [0,1]^k$ and $g : \mathcal{X} \to [0,1]^k$, by Hölder's inequality

$$\mathbb{E}(f(x) - g(x))_y = \int \min\{1, (f(x) - g(x))_y\}\,\mathrm{d}\mu(x,y)$$

$$\leq \int \min\{1, \|f(x) - g(x)\|_p\}\,\mathrm{d}\mu(x,y)$$

$$= \int \min\{1, \|f(x) - g(x)\|_p\}\frac{\mathrm{d}\mu_{\mathcal{X}}}{\mathrm{d}\nu_n}(x)\,\mathrm{d}\nu_n(x)$$

$$\leq \left\|\min\left\{1, \|f - g\|_p\right\}\right\|_{L_p(\nu_n)}\left\|\frac{\mathrm{d}\mu_{\mathcal{X}}}{\mathrm{d}\nu_n}\right\|_{L_q(\nu_n)}.$$

Once again invoking standard Rademacher complexity arguments (Shalev-Shwartz & Ben-David, 2014), with probability at least $1 - \delta$, every mapping $z \mapsto \min\{1, \|f(z) - g(z)\|_p^p\}$ where $f \in \mathcal{F}$ and $g \in \mathcal{G}$ satisfies

$$\int \min\{1, \|f(z) - g(z)\|_p^p\}\,\mathrm{d}\nu_n(z) \leq \frac{1}{m}\sum_{i=1}^m \min\{1, \|f(z_i) - g(z_i)\|_p^p\} + 3\sqrt{\frac{\ln(1/\delta)}{2m}}$$

$$+ 2\mathrm{Rad}_m\left(\left\{z \mapsto \min\{1, \|f(z) - g(z)\|_p^p\} : f \in \mathcal{F}, g \in \mathcal{G}\right\}\right).$$

Combining these bounds and unioning the two failure events gives the first bound.

For the final Rademacher complexity estimate, first note $r \mapsto \min\{1, r\}$ is 1-Lipschitz and can be peeled off, thus

$$
m\text{Rad}_m \left( \left\{ z \mapsto \min\{1, \|f(z) - g(z)\|_p^p\} : f \in \mathcal{F}, g \in \mathcal{G} \right\} \right)
$$

$$
\leq m\text{Rad}_m \left( \left\{ z \mapsto \|f(z) - g(z)\|_p^p : f \in \mathcal{F}, g \in \mathcal{G} \right\} \right)
$$

$$
= \mathbb{E}_\epsilon \sup_{\substack{f \in \mathcal{F} \\ g \in \mathcal{G}}} \sum_{i=1}^m \epsilon_i \|f(z_i) - g(z_i)\|_p^p
$$

$$
\leq \sum_{y'=1}^k \mathbb{E}_\epsilon \sup_{\substack{f \in \mathcal{F} \\ g \in \mathcal{G}}} \sum_{i=1}^m \epsilon_i |f(z_i) - g(z_i)|_{y'}^p
$$

$$
= \sum_{y'=1}^k m\text{Rad}_m \left( \left\{ z \mapsto |f(z) - g(z)|_{y'}^p : f \in \mathcal{F}, g \in \mathcal{G} \right\} \right).
$$

Since $f$ and $g$ have range $[0, 1]^k$, then $(f - g)_{y'}$ has range $[-1, 1]$ for every $y'$, and since $r \mapsto |r|^p$ is $p$-Lipschitz over $[-1, 1]$ (for any $p \in [1, \infty)$), combining this with the Lipschitz composition rule for Rademacher complexity and also the fact that a Rademacher random vector $\epsilon \in \{\pm 1\}^m$ is distributionally equivalent to its coordinate-wise negation $-\epsilon$, then, for every $y' \in [k]$,

$$
\text{Rad}_m(\{z \mapsto |f(z) - g(z)|_{y'}^p : f \in \mathcal{F}, g \in \mathcal{G}\})
$$

$$
\leq p\text{Rad}_m(\{z \mapsto (f(z) - g(z))_{y'} : f \in \mathcal{F}, g \in \mathcal{G}\})
$$

$$
= \frac{p}{m} \mathbb{E}_\epsilon \sup_{f \in \mathcal{F}} \sup_{g \in \mathcal{G}} \sum_{i=1}^m \epsilon_i (f(z_i) - g(z_i))_{y'}
$$

$$
= \frac{p}{m} \mathbb{E}_\epsilon \sup_{f \in \mathcal{F}} \sum_{i=1}^m \epsilon_i f(z_i)_{y'} + \frac{p}{m} \mathbb{E}_\epsilon \sup_{g \in \mathcal{G}} \sum_{i=1}^m -\epsilon_i g(z_i)_{y'}
$$

$$
= p\text{Rad}_m(\{z \mapsto f(z)_{y'} : f \in \mathcal{F}\}) + p\text{Rad}_m(\{z \mapsto g(z)_{y'} : g \in \mathcal{G}\}).
$$

$\square$

To prove Lemma 1.1, it still remains to collect a few convenient properties of the softmax.

**Lemma A.2.** *For any $v \in \mathbb{R}^k$ and $y \in \{1, \ldots, k\}$,*

$$
2(1 - \phi_\gamma(v))_y \geq \mathbb{1}[y \neq \arg\max_i v_i].
$$

*Moreover, for any functions $\mathcal{F}$ with $\mathcal{F} \ni f : \mathcal{X} \to \mathbb{R}^k$,*

$$
\text{Rad}_n \left( \{(x, y) \mapsto \phi_\gamma(f(x))_y : f \in \mathcal{F}\} \right) = \widetilde{\mathcal{O}} \left( \frac{\sqrt{k}}{\gamma} \text{Rad}_n(\mathcal{F}) \right).
$$

*Proof.* For the first property, let $v \in \mathbb{R}^k$ be given, and consider two cases. If $y = \arg\max_i v_i$, then $\phi_\gamma(v) \in [0, 1]^k$ implies

$$
2\left(1 - \phi_\gamma(v)\right)_y \geq 0 = \mathbb{1}[y \neq \arg\max_i v_i].
$$

On the other hand, if $y \neq \arg\max_i v_i$, then $\phi_\gamma(v)_y \leq 1/2$, and

$$
2\left(1 - \phi_\gamma(v)\right)_y \geq 1 = \mathbb{1}[y \neq \arg\max_i v_i].
$$

The second part follows from a multivariate Lipschitz composition lemma for Rademacher complexity due to (Foster & Rakhlin, 2019, Theorem 1); all that remains to prove is that $v \mapsto \phi_\gamma(v)_y$ is $(1/\gamma)$-Lipschitz with respect to the $\ell_\infty$ norm for any $v \in \mathbb{R}^k$ and $y \in [k]$. To this end, note that

$$
\frac{\mathrm{d}}{\mathrm{d}v_y} \phi_\gamma(v)_y = \frac{\exp(v/\gamma)_y \sum_{j \neq y} \exp(v/\gamma)_j}{\gamma(\sum_j \exp(v/\gamma)_j)^2}, \qquad \frac{\mathrm{d}}{\mathrm{d}v_{i \neq y}} \phi_\gamma(v)_y = -\frac{\exp(v/\gamma)_y \exp(v/\gamma)_i}{\gamma(\sum_j \exp(v/\gamma)_j)^2},
$$

and therefore

$$\left\|\nabla\phi_\gamma(v)_y\right\|_1 = \frac{2\exp(v/\gamma)_y \sum_{j\neq y}\exp(v/\gamma)_j}{\gamma(\sum_j \exp(v/\gamma)_j)^2} \leq \frac{1}{\gamma},$$

and thus, by the mean value theorem, for any $u \in \mathbb{R}^k$ and $v \in \mathbb{R}^k$, there exists $z \in [u, v]$ such that

$$\left|\phi_\gamma(v)_y - \phi_\gamma(u)_y\right| = \left|\langle\nabla\phi_\gamma(z)_y, v - u\rangle\right| \leq \|v - u\|_\infty \cdot \left\|\nabla\phi_\gamma(v)_y\right\|_1 \leq \frac{1}{\gamma}\|v - u\|_\infty,$$

and in particular $v \mapsto \phi_\gamma(v)/y$ is $(1/\gamma)$-Lipschitz with respect to the $\ell_\infty$ norm. Applying the aforementioned Lipschitz composition rule (Foster & Rakhlin, 2019, Theorem 1),

$$\mathrm{Rad}_n\left(\{(x, y) \mapsto \phi_\gamma(f(x))_y : f \in \mathcal{F}\}\right) = \widetilde{\mathcal{O}}\left(\frac{\sqrt{k}}{\gamma}\mathrm{Rad}_n(\mathcal{F})\right).$$

$\square$

Lemma 1.1 now follows by combining Lemmas A.1 and A.2.

*Proof of Lemma 1.1.* Define $\psi := 1 - \phi_\gamma$. The bound follows by instantiating Lemma A.1 with $p = 1$ and the two function classes

$$\mathcal{Q}_\mathcal{F} := \{(x, y) \mapsto \psi(f(x)_y) : f \in \mathcal{F}\} \qquad \text{and} \qquad \mathcal{Q}_\mathcal{G} := \{(x, y) \mapsto \psi(g(x)_y) : g \in \mathcal{G}\},$$

combining its simplified Rademacher upper bounds with the estimates for $\mathrm{Rad}_m(\mathcal{Q}_\mathcal{F})$ and $\mathrm{Rad}_m(\mathcal{Q}_\mathcal{G})$ and $\mathrm{Rad}_n(\mathcal{Q}_\mathcal{G})$ from Lemma A.2, and by using Lemma A.2 to lower bound the left hand side with

$$\mathbb{E}\psi(f(x))_y = \mathbb{E}(1 - \phi_\gamma(f(x))_y) \geq \frac{1}{2}\mathbb{1}\left[\arg\max_{y'} f(x)_{y'} \neq y\right],$$

and lastly noting that

$$\frac{1}{m}\sum_{i=1}^m \|\psi(f(z_i)) - \psi(g(z_i))\|_1 = \frac{1}{m}\sum_{i=1}^m \|1 - \phi_\gamma(f(z_i)) - 1 + \phi_\gamma(g(z_i))\|_1 = \Phi_{\gamma,m}(f, g).$$

$\square$

To complete the proofs for Section 1.1, it remains to handle the data augmentation error, namely the term $\|\mathrm{d}\mu_\mathcal{X}/\mathrm{d}\nu_n\|_\infty$. This proof uses the following result about Gaussian kernel density estimation.

**Lemma A.3** (See (Jiang, 2017, Theorem 2 and Remark 8)). *Suppose density $p$ is $\alpha$-Hölder continuous, meaning $|p(x) - p(x')| \leq C_\alpha\|x - x'\|^\alpha$ for some $C_\alpha \geq 0$ and $\alpha \in [0, 1]$. There there exists a constant $C \geq 0$, depending on $\alpha$, $C_\alpha$, $\max_{x\in\mathbb{R}^d} p(x)$, and the dimension, but independent of the sample size, so that with probability at least $1 - 1/n$, the Gaussian kernel density estimate with bandwidth $\sigma^2 I$ where $\sigma = n^{-1/(2\alpha+d)}$ satisfies*

$$\sup_{x\in\mathbb{R}^d} |p(x) - p_n(x)| \leq C\sqrt{\frac{\ln(n)}{n^{2\alpha/(2\alpha+d)}}}.$$

The proof of Lemma 1.2 follows.

*Proof of Lemma 1.2.* The proposed data augmentation measure $\nu_n$ has a density $p_{n,\beta}$ over $[0, 1]^d$, and it has the form

$$p_{n,\beta}(x) = \beta + (1 - \beta)p_n(x),$$

where $\beta = 1/2$, and $p_n$ is the kernel density estimator as described in Lemma A.3, whereby

$$|p_n(x) - p(x)| \leq \epsilon_n := \mathcal{O}\left(\frac{\sqrt{\ln n}}{n^{\alpha/(2\alpha+d)}}\right).$$

The proof proceeds to bound $\|\mathrm{d}\mu_\mathcal{X}/\mathrm{d}\nu_n\|_\infty = \|p/p_{n,\beta}\|_\infty$ by considering three cases.

- If $x \notin [0,1]^d$, then $p(x) = 0$ by the assumption on the support of $\mu_{\mathcal{X}}$, whereas $p_{n,\beta}(x) \geq p_n(x)/2 > 0$, thus $p(x)/p_{n,\beta}(x) = 0$.

- If $x \in [0,1]^d$ and $p(x) \geq 2\epsilon_n$, then $p_{n,\beta}(x) \geq (1-\beta)p(x) - \epsilon_n) \geq \epsilon_n/2$, and

$$
\begin{aligned}
\frac{p(x)}{p_{n,\beta}(x)} &= 1 + \frac{p(x) - p_{n,\beta}(x)}{p_{n,\beta}(x)} \\
&\leq 1 + \frac{\beta p(x)}{p_{n,\beta}(x)} + \frac{(1-\beta)|p(x) - p_n(x)|}{p_{n,\beta}(x)} \\
&\leq 1 + \frac{\beta p(x)}{(1-\beta)(p(x) - \epsilon_n)} + \frac{(1-\beta)\epsilon_n}{\epsilon_n/2} \\
&\leq 1 + \frac{\beta}{(1-\beta)(1 - \epsilon_n/p(x))} + 1 \\
&\leq 4.
\end{aligned}
$$

- If $x \in [0,1]^d$ and $p(x) < 2\epsilon_n$, since $p_{n,\beta}(x) \geq \beta = 1/2$, then

$$
\frac{p(x)}{p_{n,\beta}(x)} < \frac{2\epsilon_n}{\beta} = 4\epsilon_n.
$$

Combining these cases, $\|\mathrm{d}\mu_{\mathcal{X}}/\mathrm{d}\nu_n\|_\infty = \|p/p_{n,\beta}\|_\infty \leq \max\{4, 4\epsilon_n\} \leq 4 + 4\epsilon_n.$  $\square$

## B  REPLACING SOFTMAX WITH STANDARD MARGIN (RAMP) LOSS

The proof of Lemma 1.1 was mostly a reduction to Lemma A.1, which mainly needs bounded functions; for the Rademacher complexity estimates, the Lipschitz property of $\phi_\gamma$ was used. As such, the softmax can be replaced with the $(1/\gamma)$-Lipschitz ramp loss as is standard from margin-based generalization theory (e.g., in a multiclass version as appears in (Bartlett et al., 2017a)). Specifically, define $\mathcal{M}_\gamma : \mathbb{R}^k \to [0,1]^k$ for any coordinate $j$ as

$$
\mathcal{M}_\gamma(v)_j := \ell_\gamma(v_j - \arg\max_{y' \neq j} v_{y'}), \qquad \text{where } \ell_\gamma(z) := \begin{cases} 1 & z \leq 0, \\ 1 - \frac{z}{\gamma} & z \in (0, \gamma), \\ 0 & z \geq \gamma. \end{cases}
$$

We now have $\mathbb{1}[\arg\max_{y'} f(x)_{y'}] \leq \mathcal{M}_\gamma(f(x))_y$ without a factor of 2 as in Lemma A.2, and can plug it into the general lemma in Lemma A.1 to obtain the following corollary.

**Corollary B.1.** *Let temperature (margin!) parameter $\gamma > 0$ be given, along with sets of multiclass predictors $\mathcal{F}$ and $\mathcal{G}$. Then with probability at least $1 - 2\delta$ over an iid draw of data $((x_i, y_i))_{i=1}^n$ from $\mu$ and $(z_i)_{i=1}^n$ from $\nu_n$, every $f \in \mathcal{F}$ and $g \in \mathcal{G}$ satisfy*

$$
\begin{aligned}
\Pr_{y'}[\arg\max f(x)_{y'} \neq y] \leq{}& \left\| \frac{\mathrm{d}\mu_{\mathcal{X}}}{\mathrm{d}\nu_n} \right\|_\infty \frac{1}{m} \sum_{i=1}^m \|\mathcal{M}_\gamma(f) - \mathcal{M}_\gamma(g)\|_1 + \frac{1}{n} \sum_{i=1}^n \mathcal{M}_\gamma(g(x_i))_{y_i} \\
&+ \widetilde{\mathcal{O}}\left( \frac{k^{3/2}}{\gamma} \left\| \frac{\mathrm{d}\mu_{\mathcal{X}}}{\mathrm{d}\nu_n} \right\|_\infty \left(\mathrm{Rad}_m(\mathcal{F}) + \mathrm{Rad}_m(\mathcal{G})\right) + \frac{\sqrt{k}}{\gamma} \mathrm{Rad}_n(\mathcal{G}) \right) \\
&+ 3\sqrt{\frac{\ln(1/\delta)}{2n}} \left( 1 + \left\| \frac{\mathrm{d}\mu_{\mathcal{X}}}{\mathrm{d}\nu_n} \right\|_\infty \sqrt{\frac{n}{m}} \right).
\end{aligned}
$$

*Proof.* Overload function composition notation to sets of functions, meaning

$$
\mathcal{M}_\gamma \circ \mathcal{F} = \left\{ (x, y) \mapsto \mathcal{M}_\gamma(f(x))_y : f \in \mathcal{F} \right\}.
$$

First note that $\mathcal{M}_\gamma$ is $(2/\gamma)$-Lipschitz with respect to the $\ell_\infty$ norm, and thus, applying the multivariate Lipschitz composition lemma for Rademacher complexity (Foster & Rakhlin, 2019, Theorem 1) just as in the proof for the softmax in Lemma A.2,

$$
\mathrm{Rad}_m(\mathcal{M}_\gamma \circ \mathcal{F}) = \widetilde{\mathcal{O}}\left( \frac{2\sqrt{k}}{\gamma} \mathrm{Rad}_m(\mathcal{F}) \right),
$$

with similar bounds for $\text{Rad}_m(\mathcal{M}_\gamma \circ \mathcal{G})$ and $\text{Rad}_n(\mathcal{M}_\gamma \circ \mathcal{G})$. The desired statement now follows by combining these Rademacher complexity bounds with Lemma 1.1 applied to $\mathcal{M}_\gamma \circ \mathcal{F}$ and $\mathcal{M}_\gamma \circ \mathcal{G}$, and additionally using $\mathbb{1}[\arg\max_{y'} f(x)_{y'} \neq y] \leq \mathcal{M}_\gamma(f(x))_y$. $\qquad\square$

## C  SAMPLING TOOLS

The proofs of Lemma 3.1 and Lemma 3.2 both make heavy use of sampling.

**Lemma C.1** (Maurey (Pisier, 1980))**.** *Suppose random variable $V$ is almost surely supported on a subset $S$ of some Hilbert space, and let $(V_1, \ldots, V_k)$ be $k$ iid copies of $V$. Then there exist $(\hat{V}_1, \ldots, \hat{V}_k) \in S^k$ with*

$$\left\| \mathbb{E}V - \frac{1}{k}\sum_i \hat{V}_i \right\|_F^2 \leq \mathop{\mathbb{E}}_{V_1,\ldots,V_k} \left\| \mathbb{E}V - \frac{1}{k}\sum_i V_i \right\|_F^2 = \frac{1}{k}\left[ \mathbb{E}\|V\|_F^2 - \|\mathbb{E}V\|_F^2 \right] \leq \frac{1}{k}\mathbb{E}\|V\|_F^2 \leq \frac{1}{k}\sup_{\hat{V}\in S}\|\hat{V}\|_F^2.$$

*Proof of Lemma C.1.* The first inequality is via the probabilistic method. For the remaining inequalities, by expanding the square multiple times,

$$\mathop{\mathbb{E}}_{V_1,\ldots,V_k} \left\| \mathbb{E}V - \frac{1}{k}\sum_i V_i \right\|_F^2 \leq \mathop{\mathbb{E}}_{V_1,\ldots,V_k} \frac{1}{k^2}\left[ \sum_i \|\mathbb{E}V - V_i\|_F^2 + \sum_{i\neq j}\left\langle \mathbb{E}V - V_i, \mathbb{E}V - V_j \right\rangle \right]$$

$$= \frac{1}{k}\mathbb{E}_{V_1}\|V_1 - \mathbb{E}V\|_F^2 = \frac{1}{k}\left[ \mathbb{E}\|V\|_F^2 - \|\mathbb{E}V\|_F^2 \right] \leq \frac{1}{k}\mathbb{E}\|V\|_F^2 \leq \frac{1}{k}\sup_{\hat{V}\in S}\|\hat{V}\|_F^2.$$

$\qquad\square$

A first key application of Lemma C.1 is to sparsify products, as used in Lemma 3.2.

**Lemma C.2.** *Let matrices $A \in \mathbb{R}^{d\times m}$ and $B \in \mathbb{R}^{n\times m}$ be given, along with sampling budget $k$. Then there exists a selection $(i_1, \ldots, i_k)$ of indices and a corresponding diagonal* sampling matrix *$M$ with at most $k$ nonzero entries satisfying*

$$M := \frac{\|A\|_F^2}{k}\sum_{j=1}^k \frac{\mathbf{e}_{i_j}\mathbf{e}_{i_j}^\mathsf{T}}{\|A\mathbf{e}_{i_j}\|^2} \qquad and \qquad \left\|AB^\mathsf{T} - AMB^\mathsf{T}\right\|^2 \leq \frac{1}{k}\|A\|^2\|B\|^2.$$

*Proof of Lemma C.2.* For convenience, define columns $a_i := A\mathbf{e}_i$ and $b_i := B\mathbf{e}_i$ for $i \in \{1, \ldots, m\}$. Define *importance weighting* $\beta_i := (\|a_i\|/\|A\|_F)^2$, whereby $\sum_i \beta_i = 1$, and let $V$ be a random variable with

$$\Pr\left[ V = \beta_i^{-1}a_i b_i^\mathsf{T} \right] = \beta_i,$$

whereby

$$\mathbb{E}V = \sum_{i=1}^m \beta_i^{-1}a_i b_i^\mathsf{T}\beta_i = \sum_{i=1}^m (A\mathbf{e}_i)(B\mathbf{e}_i)^\mathsf{T} = A\left[\sum_{i=1}^m \mathbf{e}_i\mathbf{e}_i^\mathsf{T}\right]B^\mathsf{T} = A[I]B^\mathsf{T} = AB,$$

$$\mathbb{E}\|V\|^2 = \sum_{i=1}^m \beta_i^{-2}\|a_i b_i^\mathsf{T}\|_F^2\beta_i = \sum_{i=1}^m \beta_i^{-1}\|a_i\|^2\|b_i\|^2 = \sum_{i=1}^m \|A\|_F^2\|b_i\|^2 = \|A\|_F^2 \cdot \|B\|_F^2.$$

By Lemma C.1, there exist indices $(i_1, \ldots, i_k)$ and matrices $\hat{V}_j := \beta_{i_j}^{-1}a_{i_j}b_{i_j}^\mathsf{T}$ with

$$\left\| AB^\mathsf{T} - \frac{1}{k}\sum_j \hat{V}_j \right\|^2 \leq \left\| \mathbb{E}V - \frac{1}{k}\sum_j \hat{V}_j \right\|^2 = \frac{1}{k}\left[ \|A\|_F^2\|B\|_F^2 - \|AB\|_F^2 \right] \leq \frac{1}{k}\|A\|_F^2\|B\|_F^2.$$

To finish, by the definition of $M$,

$$\frac{1}{k}\sum_j \hat{V}_j = \frac{1}{k}\sum_j \beta_{i_j}^{-1}(A\mathbf{e}_{i_j})(B\mathbf{e}_{i_j})^\mathsf{T} = A\left[ \frac{1}{k}\sum_j \beta_{i_j}^{-1}\mathbf{e}_{i_j}\mathbf{e}_{i_j}^\mathsf{T} \right]B^\mathsf{T} = A[M]B^\mathsf{T}.$$

$\qquad\square$

A second is to cover the set of matrices $W$ satisfying a norm bound $\|W^\mathsf{T}\|_{2,1} \leq r$. The proof here is more succinct and explicit than the one in (Bartlett et al., 2017a, Lemma 3.2).

**Lemma C.3** (See also (Bartlett et al., 2017a, Lemma 3.2)). *Let norm bound $r \geq 0$, $X \in \mathbb{R}^{n \times d}$, and integer $k$ be given. Define a family of matrices*

$$\mathcal{M} := \left\{ \frac{r\|X\|_{\mathrm{F}}}{k} \sum_{l=1}^{k} \frac{s_l \mathbf{e}_{i_l} \mathbf{e}_{j_l}^\mathsf{T}}{\|X\mathbf{e}_{j_l}\|} : s_l \in \{\pm 1\}, i_l \in \{1,\ldots,n\}, j_l \in \{1,\ldots,d\} \right\}.$$

*Then*

$$|\mathcal{M}| \leq (2nd)^k, \qquad \sup_{\|W^\mathsf{T}\|_{2,1} \leq r} \min_{\hat{W} \in \mathcal{M}} \|WX^\mathsf{T} - \hat{W}X^\mathsf{T}\|_{\mathrm{F}}^2 \leq \frac{r^2\|X\|_{\mathrm{F}}^2}{k}.$$

*Proof.* Let $W \in \mathbb{R}^{m \times d}$ be given with $\|W^\mathsf{T}\|_{2,1} \leq r$. Define $s_{ij} := W_{ij}/|W_{ij}|$, and note

$$WX^\mathsf{T} = \sum_{i,j} \mathbf{e}_i \mathbf{e}_i^\mathsf{T} W \mathbf{e}_j \mathbf{e}_j^\mathsf{T} X^\mathsf{T} = \sum_{i,j} \mathbf{e}_i W_{ij} (X\mathbf{e}_j)^\mathsf{T} = \sum_{i,j} \underbrace{\frac{|W_{ij}|\|X\mathbf{e}_j\|_2}{r\|X\|_{\mathrm{F}}}}_{=:q_{ij}} \underbrace{\frac{r\|X\|_{\mathrm{F}} s_{ij} \mathbf{e}_i (X\mathbf{e}_j)^\mathsf{T}}{\|X\mathbf{e}_j\|}}_{=:U_{ij}}.$$

Note by Cauchy-Schwarz that

$$\sum_{i,j} q_{ij} \leq \frac{1}{r\|X\|_{\mathrm{F}}} \sum_i \sqrt{\sum_j W_{ij}^2} \|X\|_{\mathrm{F}} = \frac{\|W^\mathsf{T}\|_{2,1}\|X\|_{\mathrm{F}}}{r\|X\|_{\mathrm{F}}} \leq 1,$$

potentially with strict inequality, thus $q$ is not a probability vector. To remedy this, construct probability vector $p$ from $q$ by adding in, with equal weight, some $U_{ij}$ and its negation, so that the above summation form of $WX^\mathsf{T}$ goes through equally with $p$ and with $q$.

Now define iid random variables $(V_1, \ldots, V_k)$, where

$$\Pr[V_l = U_{ij}] = p_{ij},$$

$$\mathbb{E}V_l = \sum_{i,j} p_{ij} U_{ij} = \sum_{i,j} q_{ij} U_{ij} = WX^\mathsf{T},$$

$$\|U_{ij}\| = \left\| \frac{s_{ij}\mathbf{e}_i(X\mathbf{e}_j)}{\|X\mathbf{e}_j\|_2} \right\|_{\mathrm{F}} \cdot r\|X\|_{\mathrm{F}} = |s_{ij}| \cdot \|\mathbf{e}_i\|_2 \cdot \left\| \frac{X\mathbf{e}_j}{\|X\mathbf{e}_j\|_2} \right\|_2 \cdot r\|X\|_{\mathrm{F}} = r\|X\|_{\mathrm{F}},$$

$$\mathbb{E}\|V_l\|^2 = \sum_{i,j} p_{ij}\|U_{ij}\|^2 \leq \sum_{ij} p_{ij} r^2\|X\|_{\mathrm{F}}^2 = r^2\|X\|_{\mathrm{F}}^2.$$

By Lemma C.1, there exist $(\hat{V}_1, \ldots, \hat{V}_k) \in S^k$ with

$$\left\| WX^\mathsf{T} - \frac{1}{k}\sum_l \hat{V}_l \right\|^2 \leq \mathbb{E}\left\| \mathbb{E}V_1 - \frac{1}{k}\sum_l V_l \right\|^2 \leq \frac{1}{k}\mathbb{E}\|V_1\|^2 \leq \frac{r^2\|X\|_{\mathrm{F}}^2}{k}.$$

Furthermore, the matrices $\hat{V}_l$ have the form

$$\frac{1}{k}\sum_l \hat{V}_l = \frac{1}{k}\sum_l \frac{s_l \mathbf{e}_{i_l}(X\mathbf{e}_{j_l})^\mathsf{T}}{\|X\mathbf{e}_{j_l}\|} = \left[ \frac{1}{k}\sum_l \frac{s_l \mathbf{e}_{i_l}\mathbf{e}_{j_l}^\mathsf{T}}{\|X\mathbf{e}_{j_l}\|} \right] X^\mathsf{T} =: \hat{W}X^\mathsf{T},$$

where $\hat{W} \in \mathcal{M}$. Lastly, note $|\mathcal{M}|$ has cardinality at most $(2nd)^k$. $\qquad \square$

## D  PROOFS FOR SECTION 1.2

The bulk of this proof is devoted to establishing the Rademacher bound for computation graphs in Lemma 3.1; thereafter, as mentioned in Section 3, it suffices to plug this bound and the data augmentation bound in Lemma 1.2 into Lemma 1.1, and apply a pile of union bounds.

As mentioned in Section 3, this proof follows the scheme laid out in (Bartlett et al., 2017a), with simplifications due to the removal of "reference matrices" and some norm generality.

*Proof of Lemma 3.1.* Let cover scale $\epsilon$ and per-layer scales $(\epsilon_1, \ldots, \epsilon_L)$ be given; the proof will develop a covering number parameterized by these per-layer scales, and then optimize them to derive the final covering number in terms of $\epsilon$. From there, a Dudley integral will give the Rademacher bound.

Define $\tilde{b}_i := b_i \sqrt{n}$ for convenience. As in the statement, recursively define

$$X_0^\mathsf{T} := X^\mathsf{T}, \qquad X_i^\mathsf{T} := \sigma_i\left([W_i \Pi_i D_i \natural F_i] X_{i-1}^\mathsf{T}\right).$$

The proof will recursively construct an analogous cover via

$$\hat{X}_0^\mathsf{T} := X^\mathsf{T}, \qquad \hat{X}_i^\mathsf{T} := \sigma_i\left([\hat{W}_i \Pi_i D_i \natural F_i] \hat{X}_{i-1}^\mathsf{T}\right),$$

where the choice of $\hat{W}_i$ depends on $\hat{X}_{i-1}$, and thus the total cover cardinality will product (and not simply sum) across layers. Specifically, the cover $\mathcal{N}_i$ for $\hat{W}_i$ is given by Lemma C.3 by plugging in $\|\Pi_i D_i \hat{X}_{i-1}^\mathsf{T}\|_F \le \tilde{b}_i$, and thus it suffices to choose

$$\text{cover cardinality } k := \frac{r_i^2 \tilde{b}_i^2}{\epsilon_i^2}, \qquad \text{whereby } \min_{\hat{W}_i \in \mathcal{N}_i} \|W_i \Pi_i D_i \hat{X}_{i-1}^\mathsf{T} - \hat{W}_i \Pi_i D_i \hat{X}_{i-1}^\mathsf{T}\| \le \epsilon_i.$$

By this choice (and the cardinality estimate in Lemma C.3, the full cover $\mathcal{N}$ satisfies

$$\ln|\mathcal{N}| = \sum_i \ln|\mathcal{N}_i| \le \sum_i \frac{r_i^2 \tilde{b}_i^2}{\epsilon_i^2} \ln(2m^2).$$

To optimize the parameters $(\epsilon_1, \ldots, \epsilon_L)$, the first step is to show via induction that

$$\|X_i^\mathsf{T} - \hat{X}_i^\mathsf{T}\|_F \le \sum_{j \le i} \epsilon_j \rho_j \prod_{l=j+1}^{i} s_l \rho_l.$$

The base case is simply $\|X_0^\mathsf{T} - \hat{X}^\mathsf{T}\| = \|X^\mathsf{T} - X^\mathsf{T}\| = 0$, thus consider layer $i > 0$. Using the inductive formula for $\hat{X}_i$ and the cover guarantee on $\hat{W}_i$,

$$\begin{aligned}
\left\|X_i^\mathsf{T} - \hat{X}_i^\mathsf{T}\right\| &= \left\|\sigma_i([W_i \Pi_i D_i \natural F_i] X_{i-1}^\mathsf{T}) - \sigma_i([\hat{W}_i \Pi_i D_i \natural F_i] \hat{X}_{i-1}^\mathsf{T})\right\| \\
&\le \rho_i \left\|[W_i \Pi_i D_i \natural F_i] h X_{i-1}^\mathsf{T} - [\hat{W}_i \Pi_i D_i \natural F_i] \hat{X}_{i-1}^\mathsf{T}\right\| \\
&\le \rho_i \left\|[W_i \Pi_i D_i \natural F_i] X_{i-1}^\mathsf{T} - [W_i \Pi_i D_i \natural F_i] \hat{X}_{i-1}^\mathsf{T}\right\| + \rho_i \left\|[W_i \Pi_i D_i \natural F_i] \hat{X}_{i-1}^\mathsf{T} - [\hat{W}_i \Pi_i D_i \natural F_i] \hat{X}_{i-1}^\mathsf{T}\right\| \\
&\le \rho_i \left\|[W_i \Pi_i D_i \natural F_i]\right\|_2 \left\|X_{i-1}^\mathsf{T} - \hat{X}_{i-1}^\mathsf{T}\right\| + \rho_i \left\|[(W_i - \hat{W}_i) \Pi_i D_i \hat{X}_{i-1}^\mathsf{T} \natural (F_i - F_i) \hat{X}_{i-1}^\mathsf{T}]\right\| \\
&\le s_i \rho_i \sum_{j \le i-1} \epsilon_j \rho_j \prod_{l=j+1}^{i-1} s_l \rho_l + \rho_i \left\|(W_i - \hat{W}_i) \Pi_i D_i \hat{X}_{i-1}^\mathsf{T}\right\| \\
&\le \sum_{j \le i-1} \epsilon_j \rho_j \prod_{l=j+1}^{i} s_l \rho_l + \rho_i \epsilon_i \le \sum_{j \le i} \epsilon_j \rho_j \prod_{l=j+1}^{i} s_l \rho_l.
\end{aligned}$$

To balance $(\epsilon_1, \ldots, \epsilon_L)$, it suffices to minimize a Lagrangian corresponding to the cover size subject to an error constraint, meaning

$$L(\vec{\epsilon}, \lambda) = \sum_{i=1}^{L} \frac{\alpha_i}{\epsilon_i^2} + \lambda\left(\sum_{i=1}^{L} \epsilon_i \beta_i - \epsilon\right) \qquad \text{where } \alpha_i := r_i^2 \tilde{b}_i^2 \ln(2m^2), \quad \beta_i := \rho_i \prod_{l=i+1}^{L} s_l \rho_l,$$

whose unique critical point for $\vec{\epsilon} > 0$ implies the choice

$$\epsilon_i := \frac{1}{Z}\left(\frac{2\alpha_i}{\beta_i}\right)^{1/3} \qquad \text{where } Z := \frac{1}{\epsilon} \sum_i (2\alpha_i \beta_i^2)^{1/3},$$

whereby $\|X_L^\intercal - \hat{X}_L^\intercal\| \leq \epsilon$ automatically, and

$$\ln |\mathcal{N}| \leq Z^2 \sum_i \frac{r_i^2 \tilde{b}_i^2 \ln(2m^2)}{(2\alpha_i/\beta_i)^{2/3}}$$

$$= \frac{1}{\epsilon^2 2^{2/3}} \left[ 2 \sum_i r_i^{2/3} \tilde{b}_i^{2/3} \beta_i^{2/3} \ln(2m^2)^{1/3} \right]^2 \sum_i r_i^{2/3} \tilde{b}_i^{2/3} \ln(2m^2)^{1/3} \beta_i^{2/3}$$

$$= \frac{2^{4/3} \ln(2m^2)}{\epsilon^2} \left[ \sum_i \left( r_i \tilde{b}_i \rho_i \prod_{l=i+1}^{L} s_l \rho_l \right)^{2/3} \right]^3 =: \frac{\tau^2}{\epsilon^2},$$

as desired, with $\tau$ introduced for convenience in what is to come.

For the Rademacher complexity estimate, by a standard Dudley entropy integral (Shalev-Shwartz & Ben-David, 2014), setting $\hat{\tau} := \max\{\tau, 1/3\}$ for convenience,

$$n\mathrm{Rad}(\mathcal{G}) \leq \inf_\zeta 4\zeta\sqrt{n} + 12 \int_\zeta^{\sqrt{n}} \frac{\sqrt{\hat{\tau}}\epsilon}{\mathrm{d}} \epsilon = \inf_\zeta 4\zeta\sqrt{n} + 12\hat{\tau} \ln(\epsilon)\big|_\zeta^{\sqrt{n}} = \inf_\zeta 4\zeta\sqrt{n} + 12\hat{\tau}(\ln\sqrt{n} - \ln\zeta),$$

which is minimized at $\zeta = 3\hat{\tau}/\sqrt{n}$, whereby

$$n\mathrm{Rad}(\mathcal{G}) \leq 12\hat{\tau} + 6\hat{\tau}\ln n - 12\hat{\tau}\ln(3\hat{\tau}/\sqrt{n}) = 12\hat{\tau}(1 - \ln(3\hat{\tau})) \leq 12\hat{\tau} \leq 12\tau + 4.$$

$\qquad\qquad\qquad\qquad\qquad\qquad\qquad\qquad\qquad\qquad\qquad\qquad\qquad\qquad\qquad\qquad\square$

This now gives the proof of Theorem 1.3.

*Proof of Theorem 1.3.* With Lemma 1.1, Lemma 1.2, and Lemma 3.1 out of the way, the main work of this proof is to have an infimum over distillation network hyperparameters $(\vec{b}, \vec{r}, \vec{s})$ on the right hand side, which is accomplished by dividing these hyperparameters into countably many shells, and unioning over them.

In more detail, divide $(\vec{b}, \vec{r}, \vec{s})$ into shells as follows. Divide each $b_i$ and $r_i$ into shells of radius increasing by one, meaning meaning for example the first shell for $b_i$ has $b_i \leq 1$, and the $j$th shell has $b_i \in (j-1, j]$, and similarly for $r_i$; moreover, associate the $j$th shell with prior weight $q_j(b_i) := (j(j+1))^{-1}$, whereby $\sum_{j \geq 1} q_j(b_i) = 1$. Meanwhile, for $s_i$ use a finer grid where the first shell has $s_i \leq 1/L$, and the $j$th shell has $s_i \in ((j-1)/L, j/L)$, and again the prior weight is $q_j(s_i) = (j(j+1))^{-1}$. Lastly, given a full set of grid parameters $(\vec{b}, \vec{r}, \vec{s})$, associate prior weight $q(\vec{b}, \vec{r}, \vec{s})$ equal to the product of the individual prior weight, whereby the sum of the prior weights over the entire product grid is 1. Enumerate this grid in any way, and define failure probability $\delta(\vec{b}, \vec{r}, \vec{s}) := \delta \cdot q(\vec{b}, \vec{r}, \vec{s})$.

Next consider some fixed grid shell with parameters $(\vec{b'}, \vec{r'}, \vec{s'})$ and let $\mathcal{H}$ denote the set of networks for which these parameters form the tightest shell, meaning that for any $g \in \mathcal{H}$ with parameters $(\vec{b}, \vec{r}, \vec{s})$, then $(\vec{b'}, \vec{r'}, \vec{s'}) \leq (\vec{b}+1, \vec{r}+1, \vec{s}+1)$ component-wise. As such, by Lemma 1.1, with probability at least $1 - \delta(\vec{b'}, \vec{r'}, \vec{s'})$, each $g \in \mathcal{H}$ satisfies

$$\Pr[\arg\max_{y'} f(x)_{y'} \neq y] \leq 2 \left\| \frac{\mathrm{d}\mu_\mathcal{X}}{\mathrm{d}\nu_n} \right\|_\infty \Phi_{\gamma,m}(f, g) + \frac{2}{n} \sum_{i=1}^n (1 - \phi_\gamma(g(x_i))_{y_i})$$

$$+ \tilde{\mathcal{O}} \left( \frac{k^{3/2}}{\gamma} \left\| \frac{\mathrm{d}\mu_\mathcal{X}}{\mathrm{d}\nu_n} \right\|_\infty \left( \mathrm{Rad}_m(\mathcal{F}) + \mathrm{Rad}_m(\mathcal{H}) \right) + \frac{\sqrt{k}}{\gamma} \mathrm{Rad}_n(\mathcal{H}) \right)$$

$$+ 6\sqrt{\frac{\ln(q(\vec{b'}, \vec{r'}, \vec{s'})) + \ln(1/\delta)}{2n}} \left( 1 + \left\| \frac{\mathrm{d}\mu_\mathcal{X}}{\mathrm{d}\nu_n} \right\|_\infty \sqrt{\frac{n}{m}} \right).$$

To simplify this expression, first note by Lemma 3.1 and the construction of the shells (relying in particular on the finer grid for $s_i$ to avoid a multiplicative factor $L$) that

$$\text{Rad}_m(\mathcal{H}) = \tilde{\mathcal{O}}\left[\frac{1}{\sqrt{n}}\left(\sum_i\left[r_i'b_i'\rho_i\prod_{l=i+1}^L s_l'\rho_l\right]^{2/3}\right)^{3/2}\right]$$

$$= \tilde{\mathcal{O}}\left[\frac{1}{\sqrt{n}}\left(\sum_i\left[(r_i+1)(b_i+1)\rho_i\prod_{l=i+1}^L (s_l+1/L)\rho_l\right]^{2/3}\right)^{3/2}\right]$$

$$= \tilde{\mathcal{O}}\left[\frac{1}{\sqrt{n}}\left(\sum_i\left[r_ib_i\rho_i\prod_{l=i+1}^L s_l\rho_l\right]^{2/3}\right)^{3/2}\right],$$

and similarly for $\text{Rad}_m(\mathcal{H})$ (the only difference being $\sqrt{m}$ replaces $\sqrt{n}$). Secondly, to absorb the term $\ln(q(\vec{b}', \vec{r}', \vec{s}'))$, noting that $\ln(a) \le \ln(\gamma^2) + (a - \gamma^2)/(\gamma^2)$, and also using $\rho_i \ge 1$, then

$$\ln(q(\vec{r}', \vec{b}', \vec{s}')) = \mathcal{O}\left(\ln\prod_i(r_i+1)^2(b_i+1)^2((s_i+1)L)^2\right) = \mathcal{O}\left(L\ln L + \ln\prod_i r_i^{2/3}b_i^{2/3}s_i^{2/3}\right)$$

$$= \tilde{\mathcal{O}}\left(L + \sum_i\ln(r_i^{2/3}b_i^{2/3}) + \ln\prod_i s_i^{2/3}\right)$$

$$= \tilde{\mathcal{O}}\left(L + \ln(\gamma^2) + \frac{1}{\gamma^2}\sum_i\left[r_i^{2/3}b_i^{2/3} + \prod_{l>i} s_l^{2/3}\right]\right)$$

$$= \tilde{\mathcal{O}}\left(L + \frac{1}{\gamma^2}\sum_i\left[r_ib_i\prod_{l=i+1}^L s_l\right]^{2/3}\right)$$

$$= \tilde{\mathcal{O}}\left(L + \frac{1}{\gamma^2}\sum_i\left[r_ib_i\rho_i\prod_{l=i+1}^L s_l\rho_l\right]^{2/3}\right).$$

Together,

$$\Pr[\arg\max_{y'} f(x)_{y'} \ne y] \le 2\left\|\frac{d\mu_{\mathcal{X}}}{d\nu_n}\right\|_\infty \Phi_{\gamma,m}(f,g) + \frac{2}{n}\sum_{i=1}^n (1 - \phi_\gamma(g(x_i))_{y_i})$$

$$+ \tilde{\mathcal{O}}\left(\frac{k^{3/2}}{\gamma}\left\|\frac{d\mu_{\mathcal{X}}}{d\nu_n}\right\|_\infty \text{Rad}_m(\mathcal{F})\right) + 6\sqrt{\frac{\ln(1/\delta)}{2n}}\left(1 + \left\|\frac{d\mu_{\mathcal{X}}}{d\nu_n}\right\|_\infty\sqrt{\frac{n}{m}}\right)$$

$$+ \tilde{\mathcal{O}}\left(\frac{\sqrt{k}}{\gamma\sqrt{n}}\left(1 + k\left\|\frac{d\mu_{\mathcal{X}}}{d\nu_n}\right\|_\infty\sqrt{\frac{n}{m}}\right)\left(\sum_i\left[r_ib_i\rho_i\prod_{l=i+1}^L s_l\rho_l\right]^{2/3}\right)^{3/2}\right).$$

Since $h \in \mathcal{H}$ was arbitrary, the bound may be wrapped in $\inf_{g\in\mathcal{H}}$. Similarly, unioning bounding away the failure probability for all shells, since this particular shell was arbitrary, an infimum over shells can be added, which gives the final infimum over $(\vec{b}, \vec{r}, \vec{s})$. The last touch is to apply Lemma 1.2 to bound $\|d\mu_{\mathcal{X}}/d\nu_n\|_\infty$. $\qquad\square$

## E  PROOF OF STABLE RANK BOUND, THEOREM 1.4

The first step is to establish the sparsification lemma in Lemma 3.2, which in turn sparsifies each matrix product, cannot simply invoke Lemma C.2: pre-processing is necessary to control the element-wise magnitudes of the resulting matrix. Throughout this section, define the stable rank of a matrix $W$ as $\text{sr}(W) := \|W\|_F^2/\|W\|_2^2$ (or 0 when $W = 0$).

**Lemma E.1.** *Let matrices $A \in \mathbb{R}^{d \times m}$ and $B \in \mathbb{R}^{n \times m}$ be given, along with sampling budget $k$. Then there exists a selection $(i_1, \ldots, i_k)$ of indices and a corresponding diagonal sampling matrix $M$ with at most $k$ nonzero entries satisfying*

$$M := \sum_{j=1}^{k} \frac{Z_{i_j} \mathbf{e}_{i_j} \mathbf{e}_{i_j}^{\mathsf{T}}}{\|a_{i_j}\|} \quad \text{where} \quad Z_{i_j} \le \|A\|_{\mathrm{F}} \sqrt{\frac{m}{k}}, \qquad \text{and} \qquad \left\|AB^{\mathsf{T}} - AMB^{\mathsf{T}}\right\|^2 \le \frac{4}{k} \|A\|^2 \|B\|^2.$$

*Proof.* Let $\tau > 0$ be a parameter to be optimized later, and define a subset of indices $S := \{i \in \{1, \ldots, m\} : \|A\mathbf{e}_i\| \ge \tau\}$, with $S^c := \{1, \ldots, m\} \setminus S$. Let $A_\tau$ denote the matrix obtained by zeroing out columns not in $S$, meaning

$$A_\tau := \sum_{i \in S} (A\mathbf{e}_i)\mathbf{e}_i^{\mathsf{T}},$$

whereby

$$\|AB^{\mathsf{T}} - A_\tau B^{\mathsf{T}}\|_{\mathrm{F}} \le \|A - A_\tau\| \cdot \|B\| \le \|B\| \sqrt{\sum_{i \in S^c} \|A\mathbf{e}_i\|^2} \le \tau\sqrt{m}\|B\|.$$

Applying Lemma C.2 to $A_\tau B^{\mathsf{T}}$ gives

$$M := \frac{\|A_\tau\|^2}{k} \sum_{j=1}^{k} \frac{\mathbf{e}_{i_j}\mathbf{e}_{i_j}^{\mathsf{T}}}{\|A_\tau \mathbf{e}_{i_j}\|^2} = \sum_{j=1}^{k} \frac{Z_{i_j}\mathbf{e}_{i_j}\mathbf{e}_{i_j}^{\mathsf{T}}}{\|A_\tau \mathbf{e}_{i_j}\|} \quad \text{such that} \quad \|A_\tau B^{\mathsf{T}} - A_\tau M B^{\mathsf{T}}\|^2 \le \frac{1}{k}\|A_\tau\|^2\|B\|^2,$$

where $Z_{i_j}$ is specified by these equalities. To simplify, note $\|A_\tau\| \le \|A\|$, and $A_\tau M = AM$. Combining the two inequalities,

$$\|AB^{\mathsf{T}} - AMB^{\mathsf{T}}\| \le \|AB^{\mathsf{T}} - A_\tau B^{\mathsf{T}}\| + \|A_\tau B^{\mathsf{T}} - A_\tau M B^{\mathsf{T}}\| \le \tau\sqrt{m}\|B\| + \frac{1}{\sqrt{k}}\|A\|\|B\|.$$

To finish, setting $\tau := \|A\|/\sqrt{mk}$ gives the bound, and ensures that the scaling term $Z_{i_j}$ satisfies, for any $i_j \in S$,

$$Z_{i_j} = \frac{\|A_\tau\|^2}{k\|A_\tau \mathbf{e}_{i_j}\|} \le \frac{\|A\|_{\mathrm{F}}^2}{k\tau} = \|A\|_{\mathrm{F}} \sqrt{\frac{m}{k}}.$$

$\square$

With this tool in hand, the proof of Lemma 3.2 is as follows.

*Proof of Lemma 3.2.* Let $X_j$ denote the network output after layer $j$, meaning

$$X_0^{\mathsf{T}} := X^{\mathsf{T}}, \qquad X_j^{\mathsf{T}} := \sigma_j(W_j X_{j-1}^{\mathsf{T}}),$$

whereby

$$\|X_j^{\mathsf{T}}\|_{\mathrm{F}} = \|\sigma_j(W_j X_{j-1}^{\mathsf{T}}) - \sigma_j(0)\|_{\mathrm{F}} \le \|W_j X_{j-1}^{\mathsf{T}}\|_{\mathrm{F}} \le \|W_j\|_2 \|X_{j-1}^{\mathsf{T}}\|_{\mathrm{F}} \le \|X\|_{\mathrm{F}} \prod_{i \le j} \|W_i\|_2.$$

The proof will inductively choose sampling matrices $(M_1, \ldots, M_L)$ as in the statement and construct

$$\hat{X}_0^{\mathsf{T}} := X^{\mathsf{T}}, \qquad \hat{X}_j^{\mathsf{T}} := \Pi_j \sigma_j(W_j M_j \hat{X}_{j-1}^{\mathsf{T}}),$$

where $\Pi_j$ denotes projection onto the Frobenius-norm ball of radius $\|X\|_{\mathrm{F}} \prod_{i \le j} \|W_i\|_2$ (whereby $\Pi_j X_j = X_j$), satisfying

$$\left\|X_j - \hat{X}_j\right\|_{\mathrm{F}} \le \|X\|_{\mathrm{F}} \left[\prod_{p=1}^{j} \|W_p\|_2\right] \sum_{i=1}^{j} \sqrt{\frac{\mathrm{sr}(W_i)}{k_i}},$$

which gives the desired bound after plugging in $j = L$.

Proceeding with the inductive construction, the base case is direct since $\hat{X}_0 = X = X_0$ and $\left\| X_0 - \hat{X}_0 \right\|_{\mathrm{F}} = 0$, thus consider some $j > 0$. Applying Lemma E.1 to the matrix multiplication $W_j \hat{X}_{j-1}$ with $k_j$ samples, there exists a multiset of $S_j$ coordinates and a corresponding sampling matrix $M_j$, as specified in the statement, satisfying

$$\left\| W_j \hat{X}_{j-1}^\mathsf{T} - W_j M_j \hat{X}_{j-1}^\mathsf{T} \right\|_{\mathrm{F}} \leq \frac{1}{\sqrt{k_j}} \|W_j\|_{\mathrm{F}} \|\hat{X}_{j-1}\|_{\mathrm{F}} \leq \frac{1}{\sqrt{k_j}} \|W_j\|_{\mathrm{F}} \|X\|_{\mathrm{F}} \prod_{i<j} \|W_i\|_2.$$

Using the choice $\hat{X}_j^\mathsf{T} := \Pi_j \sigma_j(W_j M_j \hat{X}_{j-1}^\mathsf{T})$,

$$
\begin{aligned}
\left\| X_j - \hat{X}_j \right\|_{\mathrm{F}} &= \left\| \sigma_j(W_j X_{j-1}^\mathsf{T}) - \Pi_j \sigma_j(W_j M_j \hat{X}_{j-1}^\mathsf{T}) \right\|_{\mathrm{F}} \\
&\leq \left\| W_j X_{j-1}^\mathsf{T} - W_j M_j \hat{X}_{j-1}^\mathsf{T} \right\|_{\mathrm{F}} \\
&= \left\| W_j X_{j-1}^\mathsf{T} - W_j \hat{X}_{j-1}^\mathsf{T} + W_j \hat{X}_{j-1}^\mathsf{T} - W_j M_j \hat{X}_{j-1}^\mathsf{T} \right\|_{\mathrm{F}} \\
&\leq \left\| W_j X_{j-1}^\mathsf{T} - W_j \hat{X}_{j-1}^\mathsf{T} \right\|_{\mathrm{F}} + \left\| W_j \hat{X}_{j-1}^\mathsf{T} - W_j M_j \hat{X}_{j-1}^\mathsf{T} \right\|_{\mathrm{F}} \\
&\leq \|W_j\|_2 \left\| X_{j-1} - \hat{X}_{j-1} \right\|_{\mathrm{F}} + \frac{1}{\sqrt{k_j}} \|W_j\|_{\mathrm{F}} \|X\|_{\mathrm{F}} \prod_{i<j} \|W_i\|_2 \\
&\leq \|W_j\|_2 \left( \|X\|_{\mathrm{F}} \left[ \prod_{i<j} \|W_i\|_2 \right] \sum_{i<j} \sqrt{\frac{\mathrm{sr}(W_i)}{k_i}} \right) + \sqrt{\frac{\mathrm{sr}(W_j)}{k_j}} \|X\|_{\mathrm{F}} \prod_{i \leq j} \|W_i\|_2 \\
&\leq \|X\|_{\mathrm{F}} \left[ \prod_{i \leq j} \|W_i\|_2 \right] \sum_{i \leq j} \sqrt{\frac{\mathrm{sr}(W_i)}{k_i}}
\end{aligned}
$$

as desired. $\qquad\square$

To prove Theorem 1.4 via Lemma 3.2, the first step is a quick tool to cover matrices element-wise.

**Lemma E.2.** *Let $\mathcal{A}$ denote matrices with at most $k_2$ nonzero rows and $k_1$ nonzero columns, entries bounded in absolute value by $b$, and total number of rows and columns each at most $m$. Then there exists a cover set $\mathcal{M} \subseteq \mathcal{A}$ satisfying*

$$|\mathcal{M}| \leq m^{k_1+k_2} \left( \frac{2b\sqrt{k_1 k_2}}{\epsilon} \right)^{k_1 k_2}, \qquad \text{and} \qquad \sup_{A \in \mathcal{A}} \min_{\hat{A} \in \mathcal{M}} \|A - \hat{A}\|_{\mathrm{F}} \leq \epsilon.$$

*Proof.* Consider some fixed set of $k_2$ nonzero rows and $k_1$ nonzero columns, and let $\mathcal{M}_0$ denote the covering set obtained by gridding the $k_1 \cdot k_2$ entries at scale $\frac{\epsilon}{\sqrt{k_1 k_2}}$, whereby

$$|\mathcal{M}_0| \leq \left( \frac{2b\sqrt{k_1 k_2}}{\epsilon} \right)^{k_1 k_2}.$$

For any $A \in \mathcal{A}$ with these specific nonzero rows and columns, the $\hat{A} \in \mathcal{M}_0$ obtained by rounding each nonzero entry of $A$ to the nearest grid element gives

$$\|A - \hat{A}\|^2 = \sum_{i,j} (A_{ij} - \hat{A}_{ij})^2 \leq \sum_{i,j} \left( \frac{\epsilon}{\sqrt{k_1 k_2}} \right)^2 = \epsilon^2 \sum_{i,j} \frac{1}{k_1 k_2} = \epsilon^2.$$

The final cover $\mathcal{M}$ is now obtained by unioning copies of $\mathcal{M}_0$ for all $\binom{m}{k_1}\binom{m}{k_2} \leq m^{k_1+k_2}$ possible submatrices of size $k_2 \times k_1$. $\qquad\square$

The proof of Theorem 1.4 now carefully combines the preceding pieces.

*Proof of Theorem 1.4.* The proof proceeds in three steps, as follows.

1. A covering number is estimate for sparsified networks, as output by Lemma 3.2.

2. A covering number for general networks is computed by balancing the error terms from Lemma 3.2 and its cover computed here.

3. This covering number is plugged into a Dudley integral to obtain the desired Rademacher bound.

Proceeding with this plan, let $(\hat{X}_0^\mathsf{T}, \ldots, \hat{X}_L^\mathsf{T})$ be the layer outputs (and network input) exactly as provided by Lemma 3.2. Additionally, define diagonal matrices $D_j := \sum_{l \in S_{j+1}}^{!} \mathbf{e}_l \mathbf{e}_l^\mathsf{T}$ (with $D_L = I$, where the "!" denotes unique inclusion; these matrices capture the effect of the subsequent sparsification, and can be safely inserted after each $W_j$ without affecting $\hat{X}_j^\mathsf{T}$, meaning

$$\hat{X}_j^\mathsf{T} = \Pi_j \sigma_j(W_j M_j \hat{X}_{j-1}^\mathsf{T}) = \Pi_j \sigma_j(D_j W_j M_j \hat{X}_{j-1}^\mathsf{T}).$$

Let per-layer cover precisions $(\epsilon_1, \ldots, \epsilon_L)$ be given, which will be optimized away later. This proof will inductively construct

$$\tilde{X}_0^\mathsf{T} := X^\mathsf{T}, \qquad \tilde{X}_j^\mathsf{T} := \Pi_j \sigma_j(\tilde{W}_j \tilde{X}_{j-1}^\mathsf{T}),$$

where $\tilde{W}_j$ is a cover element for $D_j W_j M_j$, and inductively satisfying

$$\|\hat{X}_j^\mathsf{T} - \tilde{X}_j^\mathsf{T}\| \le \|X\|_\mathrm{F} m^{j/2} \sum_{i \le j} \epsilon_i \prod_{\substack{l \le j \\ l \ne i}} \|W_j\|_\mathrm{F}.$$

To construct the per-layer cover elements $\tilde{W}_j$, first note by the form of $M_j$ (and the scaling $Z_i$ provided by Lemma 3.2) that

$$b := \max_{i,l}(D_j W_j M_j)_{l,i} \le \max_i \|W_j M_j \mathbf{e}_i\| \le Z_i \frac{\|W_j \mathbf{e}_i\|}{\|W_j \mathbf{e}_i\|} \le \|W_j\|_\mathrm{F} \sqrt{\frac{m}{k_{j-1}}}.$$

Consequently, by Lemma E.2, there exists a cover $\mathcal{C}_j$ of matrices of the form $D_j W_j M_j$ satisfying

$$|\mathcal{C}_j| \le m^{k_j + k_{j-1}} \left( \frac{2b\sqrt{k_j k_{j-1}}}{\epsilon_j} \right)^{k_j k_{j-1}} \le m^{k_j + k_{j-1}} \left( \frac{2\|W_j\|_\mathrm{F} \sqrt{k_j m}}{\epsilon_j} \right)^{k_j k_{j-1}},$$

and the closest cover element $\tilde{W}_j \mathcal{C}_j$ to $D_j W_j M_j$ satisfies $\|D_j W_j M_j - \tilde{W}_j\|_\mathrm{F} \le \epsilon_j$.

Proceeding with the induction, the base case has $\|\hat{X}_0^\mathsf{T} - \tilde{X}_0^\mathsf{T}\| = \|X^\mathsf{T} - X^\mathsf{T}\| = 0$, thus consider $j > 0$. The first step is to estimate the spectral norm of $D_j W_j M_j$, which can be coarsely upper bounded via

$$\|D_j W_j M_j\|_2^2 \le \|D_j W_j M_j\|_\mathrm{F}^2 \le \sum_i \|W_j M_j \mathbf{e}_i\|^2 \le \sum_i \|W_j\|_\mathrm{F}^2 \frac{m}{k_{j-1}} \le \|W_j\|_\mathrm{F}^2 m.$$

By the form of $\hat{X}_j$ and $\tilde{X}_j$,

$$\begin{aligned}
\|\hat{X}_j^\mathsf{T} - \tilde{X}_j^\mathsf{T}\| &= \|\Pi_j \sigma_j(D_j W_j M_j \hat{X}_{j-1}^\mathsf{T}) - \Pi_j \sigma_j(\tilde{W}_j \tilde{X}_{j-1}^\mathsf{T})\| \\
&\le \|D_j W_j M_j \hat{X}_{j-1}^\mathsf{T} - \tilde{W}_j \tilde{X}_{j-1}^\mathsf{T}\| \\
&\le \|D_j W_j M_j \hat{X}_{j-1}^\mathsf{T} - D_j W_j M_j \tilde{X}_{j-1}^\mathsf{T}\| + \|D_j W_j M_j \tilde{X}_{j-1}^\mathsf{T} - \tilde{W}_j \tilde{X}_{j-1}^\mathsf{T}\| \\
&\le \|D_j W_j M_j\|_2 \|\hat{X}_{j-1}^\mathsf{T} - \tilde{X}_{j-1}^\mathsf{T}\|_\mathrm{F} + \|D_j W_j M_j - \tilde{W}_j\|_2 \|\tilde{X}_{j-1}^\mathsf{T}\|_\mathrm{F} \\
&\le \sqrt{m} \|W_j\|_\mathrm{F} \left[ \|X\|_\mathrm{F} m^{(j-1)/2} \sum_{i < j} \epsilon_i \prod_{\substack{l < j \\ l \ne i}} \|W_j\|_\mathrm{F} \right] + \epsilon_j \|X\|_\mathrm{F} \prod_{i < j} \|W_j\|_2 \\
&\le \|X\|_\mathrm{F} m^{j/2} \sum_{i \le j} \epsilon_i \prod_{\substack{l \le j \\ l \ne i}} \|W_j\|_\mathrm{F},
\end{aligned}$$

which establishes the desired bound on the error.

The next step is to optimize $k_j$. Let $\epsilon > 0$ be arbitrary, and set $\epsilon_j^{-1} := \epsilon^{-1} 2L\sqrt{m} \|X\|_{\text{F}} \prod_{i \neq j} \|W_i\|_{\text{F}}$, whereby

$$\|\hat{X}_L^{\mathsf{T}} - \tilde{X}_L^{\mathsf{T}}\|_{\text{F}} \leq \frac{\epsilon}{2}, \qquad |\mathcal{C}_j| \leq m^{k_j + k_{j-1}} \left( \frac{4mL\sqrt{k_j}\|X\|_{\text{F}} \prod_i \|W_i\|_{\text{F}}}{\epsilon} \right)^{k_j k_{j-1}}.$$

The overall network cover $\mathcal{N}$ is the product of the covers for all layers, and thus has cardinality satisfying

$$\ln|\mathcal{N}| \leq \sum_j \ln|\mathcal{C}_j| \leq 2\sum_j k_j \ln m + \sum_j k_j k_{j-1} \ln \left( \frac{4mL\sqrt{k_j}\|X\|_{\text{F}} \prod_i \|W_i\|_{\text{F}}}{\epsilon_j} \right)$$

$$\leq 2\sum_j k_j \ln m + \sum_j 2k_j^2 \ln \left( \frac{4mL\sqrt{k_j}\|X\|_{\text{F}}}{\epsilon_j} \right) + \left[ \sum_j 2k_j^2 \right] \cdot \left[ \sum_j \ln \|W_j\|_{\text{F}} \right].$$

To choose $(k_1, \ldots, k_L)$, letting $X_L^{\mathsf{T}}$ denote the output of the original unsparsified network, note firstly that the full error bound satisfies

$$\|X_L^{\mathsf{T}} - \tilde{X}_L^{\mathsf{T}}\| \leq \|X_L^{\mathsf{T}} - \hat{X}_L^{\mathsf{T}}\| + \|\hat{X}_L^{\mathsf{T}} - \tilde{X}_L^{\mathsf{T}}\|$$

$$\leq \sum_i \frac{\alpha_i}{\sqrt{k_i}} + \frac{\epsilon}{2}, \qquad \text{where } \alpha_i := \|X\|_{\text{F}} \left[ \prod_i \|W_i\|_2 \right] \sqrt{\text{sr}(W_i)}.$$

To choose $k_i$, the approach here is to minimize a Lagrangian corresponding to the cover cardinality, subject to the total cover error being $\epsilon$. Simplifying the previous expressions and noting $2k_j k_{j-1} \leq k_j^2 + k_{j-1}^2$, whereby the dominant term in $\ln|\mathcal{N}|$ is $\sum_j k_j^2$, consider Lagrangian

$$L(k_1, \ldots, k_l, \lambda) := \sum_i k_i^2 + \lambda \left( \sum_i \frac{\alpha_i}{\sqrt{k_i}} - \frac{\epsilon}{2} \right),$$

which has critical points when each $k_i$ satisfies

$$\frac{k_i^{5/2}}{\alpha_i} = \frac{\lambda}{4},$$

thus $k_i := \alpha_i^{2/5}/Z$ with $Z := \epsilon^2/(2\sum_j \alpha_j^{4/5})^2$. As a sanity check (since it was baked into the Lagrangian), plugging this into the cover error indeed gives

$$\|X_L^{\mathsf{T}} - \tilde{X}_L^{\mathsf{T}}\| \leq \sum_i \frac{\alpha_i}{\sqrt{k_i}} + \frac{\epsilon}{2} = \sqrt{Z} \sum_i \alpha_i^{4/5} + \frac{\epsilon}{2} = \epsilon.$$

To upper bound the cover cardinality, first note that

$$\sum_i k_i^2 = \frac{1}{Z^2} \sum_i \alpha_i^{4/5} = \frac{4}{\epsilon^4} \left( \sum_i \alpha_i^{4/5} \right)^5,$$

whereby

$$\ln|\mathcal{N}| = \widetilde{\mathcal{O}} \left( \left[ \sum_i k_i^2 \right] \cdot \left[ \sum_i \ln \|W_i\|_{\text{F}} \right] \right)$$

$$= \frac{\beta}{\epsilon^4} \qquad \text{where } \beta = \widetilde{\mathcal{O}} \left( \|X\|_{\text{F}}^4 \left[ \prod_j \|W_j\|_2^4 \right] \left[ \sum_i \text{sr}(W_i)^{2/5} \right]^5 \left[ \sum_i \ln \|W_i\|_{\text{F}} \right] \right).$$

The final step is to apply a Dudley entropy integral (Shalev-Shwartz & Ben-David, 2014), which gives

$$n\text{Rad}(\mathcal{F}) = \inf_\zeta \left( 4\zeta\sqrt{n} + 12 \int_\zeta^{\sqrt{n}} \frac{\sqrt{\beta}}{\epsilon^2} \, d\epsilon \right) = \inf_\zeta \left( 4\zeta\sqrt{n} + 12 \left[ \frac{1}{\zeta} - \frac{1}{\sqrt{n}} \right] \sqrt{\beta} \right).$$

Dropping the negative term gives an expression of the form $a\zeta + b/\zeta$, which is convex in $\zeta > 0$ and has critical point at $\zeta^2 = b/a$, which after plugging back in gives an upper bound $2\sqrt{ab}$, meaning

$$n\mathrm{Rad}(\mathcal{F}) \le 2\left(4\sqrt{n} \cdot 12\sqrt{\beta}\right)^{1/2} = 8\sqrt{3}n^{1/4}\beta^{1/4}.$$

Dividing by $n$ and expanding the definition of $\beta$ gives the final Rademacher complexity bound. $\quad\square$

