# OpenReview forum: "Generalization bounds via distillation"
_ICLR.cc/2021/Conference — ICLR 2021 Spotlight_

### Official Review · AnonReviewer3 · 2020-10-23
**Interesting approach to generalization bounds for deep neural networks.**

**Rating:** 8
**Confidence:** 5

**Review:**

The paper provides generalization bounds for seemingly complex neural networks
on the basis of much simpler ones. That is a good idea and something that is currently very relevant I think and the approach seems to be the natural one to take.

Essentially the bounds proved, bound the out of sample error
with a form of in sample error, average difference in predictions between complex and simple network, and complexity term for the simple network in terms of Rademacher complexity.

The authors provide a general framework which can be applied and show a particular way of using it and use recent results (Bartlett et al) to provide interesting application of the framework.


In terms of the actual bound achieved there are a few things I feel should be discussed more.

In Lemma 1.1.
The in-sample error. First it is not the in-sample classification error but the sort of a margin error that essentially is never zero for any prediction. Usually margin errors have a
linear penalty on the wrong side of the margin and zero on the correct side of the margin.
Second, there is a factor 2 in front of it. Normally, and in uniform convergence bounds, there are no constants in front of the in-sample error. (The other constants are of no concern).

The authors state that their work can be applied to  generalization bound of Arora et a. 18 that only worked for a compressed network but not the original one. Given the above comments about the actual bound achievable it is not clear to me what exactly one would get out of "distilling" the construction in Arora et al, but it seems it does not become the same bound as for the compressed network as shown by Arora et al. Comments on that would be appreciated.

Another small question: What does the early distillation phase on page 4 means (below lemma 1.3)?


I like the experimental setup, particularly using gradient descent to try and find a network to distill to.

Overall, I think the paper is well written including the proof sketches that make me believe the statements are actually provable (I did not rigorously check)

Overall, I think this is an interesting paper and should be accepted.

---

> ### Author Response · Authors · 2020-11-16
> **Thank you.**
>
>
> We thank the reviewer for their comments and time.  As mentioned in the general comments, so far our revisions are limited to a new appendix E, which we will incorporate into the paper body before the feedback period closes.
>
> - [ In the first version of this response, we asked here about the "sort of a margin error" comment, but we have now moved this below. ]
>
> - Regarding the "in-sample error" comment, we feel this is an important point and have a few responses which we will incorporate into the paper. Firstly, our most basic theoretical tool, Lemma 3.1 which is used to prove Lemma 1.1, can be combined with a standard margin loss to obtain an analog to Lemma 1.1 but in terms of standard margin terms: we have typed this up explicitly in Proposition E.2 in the new section E.3. This margin version does not have the coefficient "2", but now that we have the point of comparison, we argue the "2" is correct for the softmax loss used in our distillation bounds. Recall that the ramp loss could be inserted because it upper bounds the misclassification loss.  Similarly $2(1-\\phi_{\\gamma}(\\cdot))$ upper bounds the misclassification loss: for a quick check, note that it is $1/2$ at $0$ without doubling, and that Lemma A.1 establishes that doubling suffices in the multiclass case.
>
>   Regarding the related comment "sort of a margin error that essentially is never zero for any prediction", one potential benefit of the softmax as compare with the strict linear penalty ramp loss is we believe it should be more amenable to a gradient-based distillation procedure, precisely due to never having fully zero values or derivatives. We have not tried such an experiment, however, and could try it if the reviewer has further interest in this point of comparison.  [ Note that in an earlier version of this reply, we asked for further clarification on this point, but feel we understand it now and no longer need that specific clarification --- apologies for the edits. ]
>
> - Regarding the mention of Arora et al., we will incorporate the following more detailed description.  We can view their paper as giving firstly a compression procedure to obtain a simpler network $g$ from an initial network $f$, and secondly as producing a generalization bound for $g$.  Our claim (using our terminology in colored boxes on our first page) is that we can plug their generalization bound into our (green) "distillation complexity term", then compute our (red) "training error" and (blue) "distillation error" on their "$f$" and "$g$", and obtain a (gray) test error bound for the original network $f$, something they did not do in their paper (see the remarks below Theorem 2.1). To be clear, we are not using our Rademacher bounds or distillation procedure, only our Lemma 1.1.  In this way, our work can complement theirs, and handles a part of the picture which they left open.
>
> - Regarding "What does the early distillation phase on page 4 means": indeed this had not been explained anywhere, but now can be seen in the new Figure 4 in appendix E.1. Specifically, the bound from Lemma 1.3 can start out far smaller than the other bounds, although this degrades over time.  (Concretely, the "early distillation phase" means the early part of our procedure, with low regularization values.) While it is possible there is some other regularization procedure which continues to keep this bound smaller than the others, we feel that the $\\|X\\|_{\\textrm{F}}/n^{3/4}$ term in the bound will eventually force it to be worse than the others, and moreover we witnessed this behavior repeatedly during experiments.
>
> We very much appreciate the detailed reading provided by the reviewer, and are eager to receive further feedback and further improve our manuscript.

---

> > ### Comment · AnonReviewer3 · 2020-11-19
> > **Thank you for the detailed reply**
> >
> > I appreciate the reply.  It settles all my questions i think.
> > The part with the softmax in the bound makes good sense as a smooth function to optimize.
> > I do not need to see extra experiments but i appreciate the offer :).
> > I also appreciate  the possibility of replacing it with other loss functions, like margin loss, for comparison with existing works.
> >
> > For the paper by Arora et al. comparison, it is nice that you are able to handle the part of generalization for the original network in such a clean way, including the option to change the in sample error loss function to margin loss or similar.

---

> > > ### Author Response · Authors · 2020-11-24
> > > **Thank you.**
> > >
> > > Thank you for the kind words and for going over our lengthy reply.
> > >
> > > We have made some minor revisions to the body of the paper, incorporating discussion of the "2", margin bounds, and detailed remarks on Arora et al.; it was logical to include these all in subsection 3.1.
> > >
> > > Thank you once again for pointing these issues out; we look forward to further comments, and will gladly incorporate them!

---

### Official Review · AnonReviewer2 · 2020-10-28
**Good paper**

**Rating:** 7
**Confidence:** 2

**Review:**

The paper overall is of good quality. The story of the work is well-written which makes the contributions easier to digest. One suggestion would be to comment a bit more on the relevance of the margin distribution for readers that are unfamiliar with it, for instance, in Figure 1, the term margin distribution is thrown without explaining why one should look into it.

The topic of this work is of great significance given that understanding the generalization error of neural nets is relevant to the community.

In terms of theoretical contributions, the results seem sound although I did not check the proofs in much detail. Most of the proofs are based on known techniques to find upper bounds. Authors also show some experiments on the behavior of their bounds, and in particular, on margin distributions.

Overall, I like the idea of bounding complex models by distilled models (simpler ones), and as authors point out, to make this approach work one needs to have a solid way to find such distilled models. If my understanding is correct, the "weak" part of the work is that authors do rely on sparsification to find distilled models. While valid, my intuition is that such distillation process could still lead to large bounds. Which seem to be the case of Figure 3(b)?
To give an example, consider a task where an overparametrized complex network generalizes well in practice, it seems unlikely to me that a (very) sparse or pruned network would "perform similarly". That is, the "sparsified" model to be "close in performance" to the complex model would not be simple enough to give a non-vacuous upper bound. I would appreciate if authors can comment on this concern.

---

> ### Author Response · Authors · 2020-11-16
> **Thank you.**
>
>
> We thank the reviewer for their comments and time.  As mentioned in the general comments, so far our revisions are limited to a new appendix E, which we will incorporate into the paper body before the feedback period closes.
>
> - Regarding "relevance of the margin distribution for readers that are unfamiliar with it", we thank the reviewer and will incorporate an explanation when we roll our changes into the body at the end of the feedback period.
>
> - Regarding "the 'weak' part of the work is that authors do rely on sparsification", we have two responses.
>
>   Firstly, a clarification: we do not use sparsification in our experiments, instead we use the regularization-based procedure for all experiments, as detailed at the start of section 2.  In response to comments from AnonReviewer1, we have created a new figure, namely figure 4 in section E.1, which better illustrates this procedure, and shows how bounds improve as the procedure iterates; we hope this resolves some concerns.  Regarding the theory, it also does not rely on sparsification, and can be combined with any distillation technique and Rademacher complexity bound.  In section 1.2, we indeed gave a sparsification perspective and bound, but this was not tied to our other theory or experiments.  In response to AnonReviewer4, we have produce a new bound, Proposition E.1, which gives another setting when distillation is possible (but is not tied to sparsification), and plan to include this in section 1.2.
>
>   Secondly, following this clarification, the reviewer may still feel concerned that the method can not improve bounds for many networks (without destroying their predictive performance, equivalently requiring $\Phi(f,g)$ to be large). In response to this, we point again to figure 4 in (the new) section E.1, which shows that the distillation procedure successfully reduces various bounds by orders of magnitude before the error significantly increases.
>
>   Regarding Figure 3(b), the "block deletion" there was was a byproduct of the regularization procedure, and not any discrete deletion/sparsification procedure.
>
> - Regarding "Most of the proofs are based on known techniques to find upper bounds", we wish to highlight three places where our proofs differ significantly from prior work:
>
>     1. While our core Lemma 3.1 (powering Lemma 1.1) has connections to existing work, we do not know of a proof technique or result very similar to it in the literature.
>
>     2. We do not know of a Rademacher complexity estimate for deep networks which is similar in proof or statement to Lemma 1.3, which proceeded by sparsifying within the generalization bound.
>
>     3. While it is essentially notation and setup, our computation graph formalism allows us to tightly handle skip connections and convolution layers.
>
> We look forward to further comments and questions from the reviewer.

---

### Official Review · AnonReviewer4 · 2020-10-29
**some comments**

**Rating:** 6
**Confidence:** 2

**Review:**

The generalization performance of learning algorithms characterizes their ability to generalize their empirical behavior on training examples to unseen test data, which provides an intuitive understanding of how different parameters affect the learning performance and some guides to design learning machines. Different from the traditional error analysis, this paper focuses on bounding the divergence bettween the test error and the training error by the the corresponding distillation error and distillation complexity, e.g., test error  is bounded by training error + distillation error + distillation complexity. The current learning theory analysis may be important to understand the theoretical foundations of distillation strategy in deep networks. However, some theoretical issues should be illustrated to improve its readability, e.g,.
1)What is the relation between the original network complexity and the corresponding distillation error +distillation complexity?
2)Is the derived upper bound (e.g., Theorem 1.2) tighter than the traditional one? Please present a table to compare it with  the related error bounds.
3)What are necessary/sufficient theoretical conditions for the effectiveness of distillation strategy (from the generalization error bounds)?
4)It may be better to state some discussions for the lower bound on the generalization error. Does it also relate with distillation complexity?

---

> ### Author Response · Authors · 2020-11-16
> **Thank you.**
>
>
> We thank the reviewer for their comments and time.  As mentioned in the general comments, so far our revisions are limited to a new appendix E, which we will incorporate into the paper body before the feedback period closes.
>
> - Regarding "1.", we have included a new figure 4 in a new appendix E.1 which shows various bounds as we run our regularization-based distillation procedure. Here it can be seen that our procedure reduces bounds by orders of magnitude, even before the other terms of the bound start to become large. Does this figure satisfactorily resolve this concern?
>
> - Regarding "2.", we hope the preceding new figure 4 also clarifies this point, but appreciate comments otherwise!
>
> - Regarding "3.", for the "sufficient" part, we do not understand fully when distillation is possible, however we have produced a new bound in Proposition E.1 of the new appendix E.2 which we hope at least highlights some favorable situations, and reveals some of our intuition.  Specifically, our intuition is that one situation where distillation works well is when there are many needless values in the weight matrices which in fact cancel, and while they make the weights large, do not affect predictions.  In Proposition E.1, we have modeled this as randomness, and proved that distillation works, meaning we have a distilled $g$ of vastly smaller complexity than the original $f$, and additionally $\Phi_{1,n}(f,g)$ is small. To be clear, it is not our intent to argue that this is precisely what distillation is doing, but merely to scratch the surface and give one setting that can be analyzed.
>
>   Regarding the "necessary" part of your comment, unfortunately we can not say much at this time, as it is related to the necessary conditions of when deep networks can generalize and not, which so far require simplified settings (e.g., ones which are close to linear, at least in some sense).
>
> - Regarding "4.", this also ties to our lower bound comment in "3.", where we currently do not have much to say.  However, since we can now edit an additional ninth body page of the submission, we can add some open problems, including discussion of exactly this question.
>
> We very much appreciate the concrete feedback, and are eager to hear more from the reviewer!

---

### Official Review · AnonReviewer1 · 2020-10-29
**An interesting theoretical study on the generalization bound, while comparison between existing bounds may make it more convincing**

**Rating:** 6
**Confidence:** 2

**Review:**

This paper provides an upper boundary of the generalization error of networks: the sum of its training error, the distillation error, and the complexity of the distilled network. Then, it also provides an instantiation of the lemma applicable to residual networks, an explicit compression analysis via pruning with a corresponding generalization bound, and empirical supports.

**Pros**

* The idea of applying distillation to tackle the generalization dilemma seems interesting, and the proof seems rigorous.
* Experiments in section 2 on the width independence, depth, and random label are reasonable for me.

**Cons**

* A comparison (e.g. on the tightness of the bound, or the correlation with generalization) between the proposed distillation-based generalization bound and existing generalization bounds in literature may help demonstrating the effectiveness of the proposed bound.

---

> ### Author Response · Authors · 2020-11-16
> **Thank you.**
>
>
> We thank the reviewer for their comments and time.  As mentioned in the general comments, so far our revisions are limited to a new appendix E, which we will incorporate into the paper body before the feedback period closes.
>
> In response to the reviewer's "Con", we have produced figure 4 in section E.1, showing how the regularization-based distillation procedure can be coupled with essentially any standard generalization bound, and shrink it.  We included a detailed description there, but the main points are as follows.
>
> - The plot has the regularization coefficient as the horizontal axis, and tracks various bounds ("distillation complexities") and quantities such as test error and $\Phi$ as the distillation procedure proceeds; we hope that in addition to resolving the reviewer's concern, this figure can be used to help readers understand our distillation procedure.
>
> - Our tools are not tied to any single distillation complexity measure, but can be combined with any bound.  In this sense, our tools complement many generalization bounds.
>
> - A key baseline in Figure 4 is the VC dimension, which was identified in Figure 4 of Arora et al as a difficult baseline.  Our distillation procedure eventually brings the various bounds below it.
>
> We would like to incorporate this figure in the paper body, and are eager to hear further comments!

---

### Author Response · Authors · 2020-11-16
**Revision summary.**


We thank the reviewers for their time.

We have responded to reviewers individually; here we will only mention manuscript revisions.

For now, other than some minor cosmetic changes, we have only added an appendix E with new material, in response to reviewer feedback.  Our plan is to incorporate this material into the paper body just before the feedback period completes.  For now, the new material is as follows.

- A new figure, in the new section E.1, showing a single run of our regularization-based distillation procedure, along with plots of various bounds on distillation complexity, and various related quantities such as distillation error $\Phi_{1,n}$ and training and testing error. One purpose of this plot is to show that not only does this procedure result in vastly smaller bounds, moreover it can be combined with essentially generalization upper bound from the literature, and improve it.

- A proof, in section E.2, giving a setting where the original function $f$ has high complexity (bad generalization), but there exists a nearby function $g$ (meaning $\Phi(f,g)$ is small) which has low complexity (low distillation complexity and good generalization).

- A proof sketch, in section E.3, instantiating our core lemmas on a margin loss rather than the softmax loss.

---

> ### Author Response · Authors · 2020-11-24
> **Further revision summary.**
>
> We have made a few minor revisions:
>
> - We have incorporated comments on Arora et al.'s compression paper, margin distributions, and the "factor 2" in the core lemma, all of which arose in discussions thanks to AnonReviewer3.
>
> - We have extended the caption to figure 1 with a pointer to the margin distribution discussion in section 2, as suggested by AnonReviewer2.
>
> - We have made minor adjustments to appendix E, which was added during this rebuttal phase, which will still receive further adjustment and eventually be incorporated with the rest of the paper following further discussion.
>
> We thank the reviewers for their time.

---

### Author Response · Authors · 2021-04-01
**Camery ready revision summary.**

Our camera ready version contains the following revisions:

- We have re-done our experiments, most notably including a variety of plots showing explicitly the effect of increasing distillation on generalization bounds, and also including extensive experiments on cifar.  These experiments were requested by reviewers, but are also central to our story, and we had versions of them prior to submitting; thankfully, the long camera ready period allowed us to produce compelling versions.

- The original main lemma had a bug; fixing it made the bound more realistic and more comparable to our other main bound, and thus we feel the paper hangs together better than before.

- We have re-organized the paper and re-done a lot of writing to highlight these changes and other requests from reviewers (e.g., connections to margin theory, more detailed explanations of the distillation procedure, etc.).  We no longer had space for the depth independence figures (even with the ninth page), and decided to cut them entirely, as we feel they are not as effective as our new experiments.

We thank the reviewers once again for their time.

---

### Decision · Program_Chairs · 2021-01-07
**Final Decision**

**Decision:**

Accept (Spotlight)

**Comment:**

This paper provides a novel generalization bound for neural networks using knowledge distillation. In particular, they argue that

"test error <= training error + distillation error + distillation complexity" where the distillation complexity is typically much smaller than the original complexity of the neural network. This is motivated by the empirical findings that neural networks can typically be significantly compressed in practice using KD without losing too much accuracy.


I found this result novel and the direction is very promising. This is a clear accept for ICLR.